# Modeling germline mutations in pineoblastoma uncovers lysosome disruption-based therapy

Philip E. D. Chung [1,2], Deena M. A. Gendoo [3,15], Ronak Ghanbari-Azarnier[1,2,15], Jeff C. Liu[1], Zhe Jiang[1], Jennifer Tsui[1], Dong-Yu Wang [1], Xiao Xiao[1,4,5], Bryan Li [2,5], Adrian Dubuc[2,6], David Shih [2,6], Marc Remke[6], Ben Ho[6], Livia Garzia[6,7], Yaacov Ben-David[4,5], Seok-Gu Kang [8], Sidney Croul[9], Benjamin Haibe-Kains[10,11,12], Annie Huang[2,5], Michael D. Taylor [2,5,13] & Eldad Zacksenhaus[1,2,14✉]

Pineoblastoma is a rare pediatric cancer induced by germline mutations in the tumor suppressors *RB1* or *DICER1*. Presence of leptomeningeal metastases is indicative of poor prognosis. Here we report that inactivation of Rb plus p53 via a WAP-Cre transgene, commonly used to target the mammary gland during pregnancy, induces metastatic pineoblastoma resembling the human disease with 100% penetrance. A stabilizing mutation rather than deletion of p53 accelerates metastatic dissemination. Deletion of Dicer1 plus p53 via WAP-Cre also predisposes to pineoblastoma, albeit with lower penetrance. In silico analysis predicts tricyclic antidepressants such as nortriptyline as potential therapeutics for both pineoblastoma models. Nortriptyline disrupts the lysosome, leading to accumulation of nonfunctional autophagosome, cathepsin B release and pineoblastoma cell death. Nortriptyline further synergizes with the antineoplastic drug gemcitabine to effectively suppress pineoblastoma in our preclinical models, offering new modality for this lethal childhood malignancy.

[1] Toronto General Research Institute, University Health Network, 67 College Street, Toronto, ON M5G 2M1, Canada. [2] Department of Laboratory Medicine & Pathobiology, University of Toronto, Toronto, ON, Canada. [3] Centre for Computational Biology, Institute of Cancer and Genomic Sciences, University of Birmingham, Birmingham, UK. [4] The Key laboratory of Chemistry for Natural Products of Guizhou Province and Chinese Academic of Sciences, Guiyang, Guizhou 550014, China. [5] State Key Laboratory for Functions and Applications of Medicinal Plants, Guizhou Medical University, Guiyang 550025, China. [6] Developmental & Stem Cell Biology Program, The Hospital for Sick Children, Toronto, ON, Canada. [7] Faculty of Medicine, department of surgery, McGill University, Quebec, Canada. [8] Neurosurgery, Brain Tumor Center, Severance Hospital, Yonsei University College of Medicine, Seoul, Republic of Korea. [9] Department of Pathology & Laboratory Medicine, Division of Anatomical Pathology, Dalhousie University, Halifax, Canada. [10] Princess Margaret Cancer Centre, University Health Network, Toronto, ON, Canada. [11] Department of Medical Biophysics, University of Toronto, Toronto, Canada. [12] Vector Institute, and Ontario Institute For Cancer Research, Toronto, ON, Canada. [13] The Arthur and Sonia Labatt Brain Tumour Research Centre, The Hospital for Sick Children, Toronto, ON, Canada. [14] Department of Medicine, University of Toronto, Toronto, ON, Canada. [15] These authors contributed equally: Deena M. A. Gendoo, Ronak Ghanbari-Azarnier ✉email: eldad.zacksenhaus@utoronto.ca

Pineoblastoma (PB) is a rare but aggressive tumor of the pineal gland[1–4]. It affects approximately equally males and females with overall survival (OS) rate of about 54%. Survival drops to 15% in children ≤5 years of age[3], and the presence of metastases at diagnosis is indicative of exceedingly poor clinical outcome[4]. While PB affects adults, most cases occur in young children with a median latency of ~5–6 years. Early symptoms include headache, ocular disturbance, ataxia and increased intracranial pressure due to hydrocephalus. Current treatments comprise maximal tumor resection followed by radiation and chemotherapy.

About 5% of patients with bilateral retinoblastoma (RB), an ocular tumor of infancy, develop PB, termed "trilateral RB", directly implicating the tumor suppressor *RB1* in the aetiology of this disease[5]. Although new chemotherapy regimens have improved outcome, 5-year OS of trilateral RB patients is only 44%, with the presence of leptomeningeal metastases associated with reduced survival[5]. Germline mutation in *DICER1* also predisposes to PB[6–8], whereas *DROSHA* loss and *PDE4DIP* duplication are found in sporadic cases[9]. Recent molecular classification of PBs identified several subtypes with distinct oncogenic alterations and clinical/pathological features. *RB1* loss or amplification of *MYC* or the *miR17-92* cluster is observed in young children (median 1.3-1.4 years) with 5 year OS in this cohort of 28.6% and 37.5%, respectively; whereas loss-of-function alterations in genes involved in miRNA biogenesis (*DICER1*, *DROSHA*, and *DGCR8*) characterize older patients (median age 5.2–14.0 years) with five-year OS of 68–100%[10].

Rb$^{+/−}$:p53$^{−/−}$ mice develop diverge tumor types including pituitary, thyroid, lymphoma, sarcoma, islet cell, bronchial hyperplasia, retinal dysplasia, and PB[11]. A major hurdle in developing a specific model for PB has been the lack of a proper CRE deleter line. An interphotoreceptor retinoid binding protein promoter Cre transgenic line (IRBP-Cre) targets the pineal gland, but also other lineages. Indeed, IRBP-Cre mediated deletion of Rb on a p53$^{+/−}$ background led to anterior lobe tumors, pituitary tumors, other tumors or no lesions at all in some mice as well as PB in 15% of mice[12]. Here we have made the fortuitous discovery that the whey acidic protein (WAP) promoter Cre deleter line (WAP-Cre)[13], commonly used to target mammary lobuloalveolar progenitors during the estrous cycle and pregnancy, is expressed in the pineal gland in both male and female mice, and that WAP-Cre:Rb$^{flox/flox}$:p53$^{flox/flox}$ and WAP-Cre:Rb$^{flox/flox}$:p53$^{lsl\_R270H/flox}$ mice develop metastatic PBs that resemble the human disease with short latency and 100% penetrance. Disruption of Dicer1 plus p53 also induces PB, though with longer latency and reduced penetrance as observed in children. These results demonstrate the utility of the WAP-Cre transgene for modeling diverse types of PBs, and suggesting similar cell of origin for *RB1*- and *DICER1*-deficient PB. In silico analysis predicts high sensitivity of mouse PBs to FDA-approved tricyclic antidepressant drugs such as nortriptyline (NOR). NOR disrupts the lysosome, leading to inhibition of autophagic flux, cathepsin release and the demise of both mouse and human PB cells. NOR further synergizes with the antineoplastic drug gemcitabine to effectively suppress PB in vivo. Thus, our results model the two most common germline mutations, *RB1* and *DICER1*, that predispose children to PB, and uncover a new therapeutic avenue for this devastating disease.

## Results

### Rb plus p53 deletion via WAP-Cre induces metastatic PB.

In the course of studying the effect of deleting Rb in the mammary epithelium via WAP-Cre[14], we observed that WAP-Cre:Rb$^{flox/flox}$:p53$^{flox/flox}$ mice developed brain tumors with latency of 133 days and 100% penetrance (Fig. 1a). The tumors protruded

the head, and upon histological examination, were found to engulf the entire superior part of the brain, invading the cerebellum (Fig. 1b, c). These lesions were observed in WAP-Cre:Rb$^{flox/flox}$:p53$^{flox/flox}$ (henceforth referred to as Rb/p53-deleted) mice and in one of sixteen WAP-Cre:Rb$^{flox/flox}$:p53$^{flox/wt}$ mice, but not in WAP-Cre:Rb$^{flox/flox}$, WAP-Cre:p53$^{flox/flox}$, or WAP-Cre:Rb$^{flox/flox}$:Pten$^{flox/flox}$ mice[15] (Fig. 1a). Human PBs display Homer Wright and Flexner-Winsteiner rosettes[16,17], irregular, pleomorphic nuclei, and scant cytoplasm[18]. The Rb/p53-deleted PB exhibited similar histology with pleomorphic nuclei, scarce cytoplasm and rosette features (Fig. 1d). Deletion of the floxed Rb and p53 alleles, and loss of protein expression in these tumors were confirmed (Fig. 1e).

To determine the origin of these tumors, we monitored their development by magnetic resonance imaging (MRI). While human pineal gland is located in the posterior segment of the diencephalon at the center of the brain, mouse pineal gland is positioned on the dorsal part of the brain between the midbrain and the cerebral cortex[19]. MRI of a 50-day-old mouse identified a small lesion on the superior side of the midbrain, which became larger in subsequent imaging at 70 and 106 days of age in the same mouse (Fig. 1f). To further pinpoint the tissue of origin, we crossed WAP-Cre:Rb$^{flox/flox}$:p53$^{flox/flox}$ mice to mT/mG double fluorescent Cre reporter line in which all cells fluoresce tomato-red unless the mT cassette is deleted via Cre-mediated recombination, inducing expression of enhanced Green Fluorescent Protein (eGFP)[20]. WAP-Cre:Rb$^{flox/flox}$:p53$^{flox/flox}$:mT/mG mice exhibited large green-fluorescent tumors on a red-fluorescent background (Fig. 1g). Imaging of pineal glands from young mice uncovered "green" micro-tumors at as early as 18 days of age; these micro-lesions were observed in H&E stained sections and by ki67 staining, revealing highly proliferative tumor cells with similar histology as full-blown PB (Fig. 1h, Supplementary Fig. 1a). Notably, the WAP-Cre transgene was reported to be expressed in the brain[13]. However, endogenous WAP gene is expressed at background levels in the pineal gland (Supplementary Fig. 2), suggesting that expression of the WAP-Cre transgene in this gland is likely the result of ectopic activation at the integration site.

Human PBs express neuronal markers such as synaptophysin, but little or no expression of glial markers like glial fibrillary acidic protein (GFAP)[18]. In accordance Rb/p53-deleted PBs showed widespread expression of synaptophysin but not GFAP or the epithelial markers pan cytokeratin (PanCK) and epithelial membrane antigen (EMA; Fig. 1i). Like human PBs, which exhibit a proliferation index of over 36%[21], Rb/p53-deleted PBs were highly proliferative as shown by strong and widespread ki67 expression (Fig. 1j). Consistent with p53 deletion, these tumors stained negative for p53 (Fig. 1k). Occasionally, Rb/p53-deleted mice developed pituitary tumors, which were observed as small lesions in mice with end stage PB (Supplementary Fig. 1b).

Metastasis is a poor prognostic marker for PB[4]. Analysis of whole brains and spinal cords from WAP-Cre:Rb$^{flox/flox}$:p53$^{flox/flox}$:mT/mG mice revealed multiple large metastatic lesions on the leptomeningeal surface of the spine (Fig. 1l). Thus, combined deletion of Rb plus p53 via WAP-Cre induces metastatic PB with 100% penetrance and features of human PB.

### Cluster analysis of Rb/p53-deficient PB.

We next classified the mouse Rb/p53-deficient PBs using whole genome mRNA gene-set enrichment analysis (GSEA)-based principle component analysis (PCA)[22] of 12 Rb/p53-deleted PB samples in comparison to human PB, Group 3, Group 4, and SHH medulloblastoma (MB), RB, and glioblastoma (GBM) as well as mouse RB. The Rb/p53-deleted PBs clustered close to mouse RB and human PB as

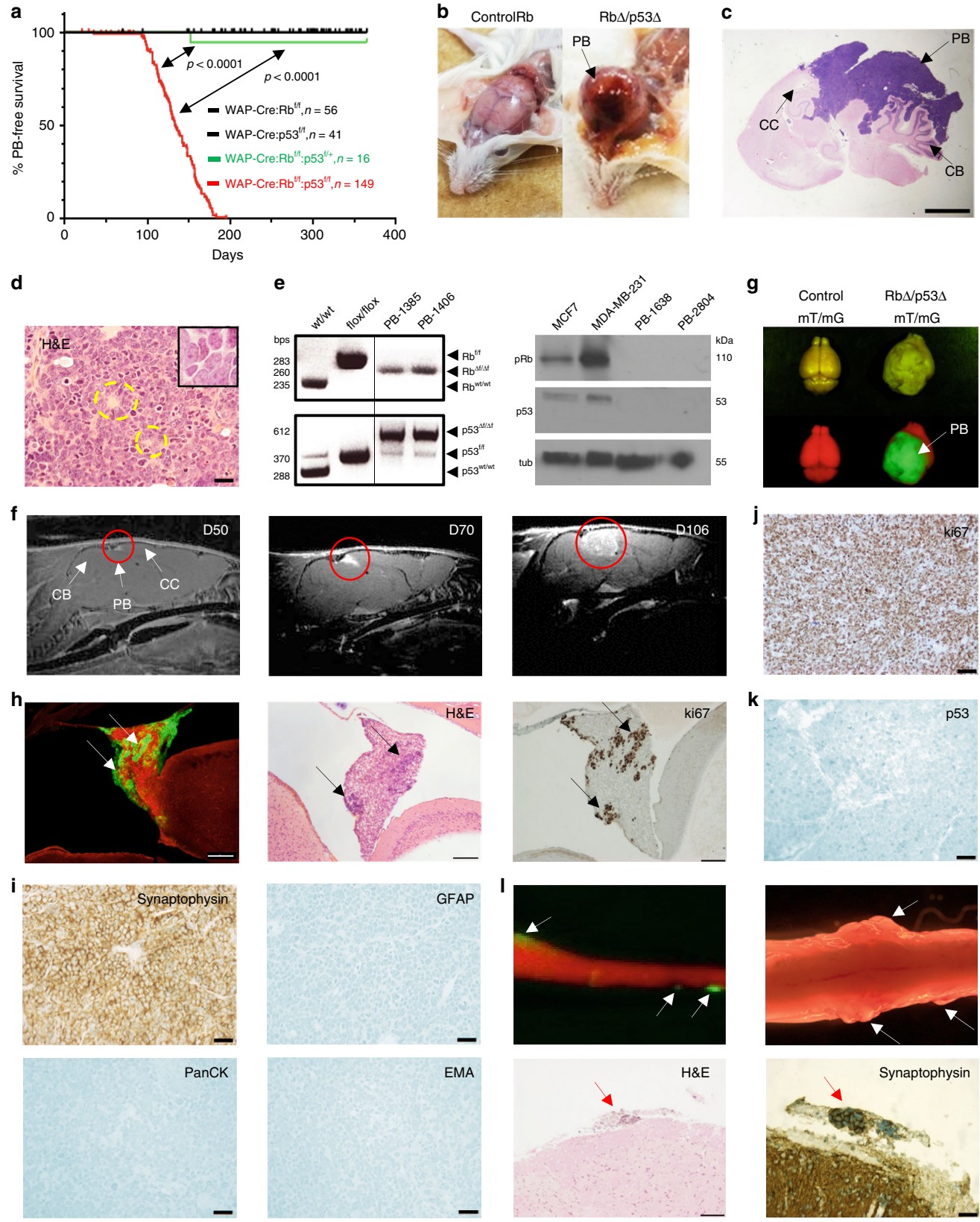

well as to Group 3 and Group 4 MBs, but relatively far from GBM as recently established for human brain tumors[23,24] (Fig. 2a, Supplementary Fig. 3).

Like group 3 MB, which expresses photoreceptor genes[25], mouse PBs exhibited high levels of *Tulp1*, *Impg2*, and *Ush2a* relative to SHH MBs (Fig. 2b). We directly compared mouse PB to

human MB using a 75-gene classifier (Fig. 2b and Supplementary Table 1)[26]. Mouse Rb/p53-deleted PBs resembled Group 3 MB with low expression of *Atoh1*, *Boc*, *Hhip*, *Cxcr4* and *Sfrp1*, and high levels of *Impg2*, *Rasgrf2*, *Ush2a*, *Cadm2*, and *Tulp1* (Fig. 2b). We then compared expression profiles of these 10 genes in human PBs versus human SHH and group 3 MBs in an

**Fig. 1 Deletion of Rb plus p53 via WAP-Cre induces pineoblastoma. a** Kaplan–Meier pineoblastoma (PB)-free survival curves for the indicated mouse models. WAP-Cre:Rb[flox/flox]:p53[flox/flox] mice (n = 149, red) developed PB with 100% penetrance and median latency of 133 days. One WAP-Cre:Rb[flox/flox]:p53[flox/+] mouse (n = 16, green) but none of WAP-Cre:Rb[flox/flox] (n = 56, black) or WAP-Cre:p53[flox/flox] (n = 41, black) mice developed PB. Statistical analysis by Mantel-Cox test. **b** Appearance of a normal control versus a WAP-Cre:Rb[flox/flox]:p53[flox/flox] mouse with a bulging PB. **c** Low magnification histology (H&E) of large PB invading the cerebellum in a WAP-Cre:Rb[flox/flox]:p53[flox/flox] mouse. Scale bar, 200 µm. CB, cerebellum; CC, cerebral cortex. **d** High magnification H&E image of PB from a WAP-Cre:Rb[flox/flox]:p53[flox/flox] mouse, showing densely packed tumor cells with irregular nuclear shape (inlet), scant cytoplasm, and rosette-like features (circled). Scale bar, 2 µm. Inlet, high power magnification (×100). **e** Left, PCR analysis of wild-type control, flox control and two tumor samples, confirming deletion of Rb and p53 in PBs. Right, immunoblots of Rb/p53-deleted PBs and control cell lines for pRb and p53. **f** MRIs of a WAP-Cre:Rb[flox/flox]:p53[flox/flox] mouse at indicated time points, showing gradual increase in PB size (red circles). **g** Dorsal images of bright field (top) and fluorescent (bottom) microscopy of control wild-type mouse (left) and PB bearing WAP-Cre:Rb[flox/flox]:p53[flox/flox]:mT/mG mice (right). Red, normal cells; green, tumor cells. **h**. Representative fluorescent and H&E images of a micro-tumor (arrows) in the pineal gland of WAP-Cre:Rb[flox/flox]:p53[flox/flox]:mT/mG mouse. Right, widespread expression of ki67 in the micro-lesion. Scale bar, 200 µm. **i** Marker analysis of PB by IHC. Synaptophysin is widely expressed; GFAP, PanCK and EMA are negative. Scale bars, 2 µm. **j, k** Representative IHC images showing strong expression of ki67 but negative expression of p53 in Rb/p53-deleted PB. Scale bar, 2 µm. **l** Representative fluorescent, bright field, H&E, and IHC images showing large metastases (arrows) on the leptomeningeal surface of the spine of WAP-Cre:Rb[flox/flox]:p53[flox/flox]:mT/mG mice. Top, original magnifications: left, ×10; right ×25. Bottom, scale bars, 2 µm. Raw data are provided in Source Data files.

independent dataset[27]; human PBs exhibited expression profiles similar to group 3 MB but not SHH MB (Fig. 2c). Thus, Rb/p53-deleted PBs are similar to both human PB and group 3 MB.

Next, we developed a 95-gene classifier that can differentiate human PB from human SHH MB (Fig. 2d; Supplementary Table 1). Using this classifier, Rb/p53-deleted PBs were again highly similar to human PB (Fig. 2d). Overall, these results indicate that mouse Rb/p53-deleted PB highly resembles human PB, and thus, WAP-Cre:Rb[flox/flox]:p53[flox/flox] mice may serve as a preclinical model for this deadly disease.

**Rb/p53-R270H PB exhibits enhanced metastasis.** In human cancer, p53 disruption involves large deletions of the gene or mutations that usually affect the DNA-binding domain, creating dominant-negative or gain of function alleles[28]. Although the status of p53 in human PB is not fully established, progression to full-blown PB is associated with high p53 immuno-staining, suggesting stabilizing p53 mutation[29]. We therefore determined the effect of expressing a p53 mutant allele, R270H, with mutation in the DNA-binding domain, on PB formation and dissemination[28]. WAP-Cre:Rb[flox/flox]:p53[lsl_R270H/flox] mice (n = 19; henceforth referred to as Rb/p53-mutated) developed PB with similar latency, penetrance, and histology compared with Rb/p53-deleted PB and closely clustered with mouse Rb/p53-deleted and human PB by GSEA-PCA (Fig. 2a; 3a–b). Tumor penetrance decreased to 4/36 in WAP-Cre:Rb[flox/flox]:p53[lsl_R270H/+] mice in which the other chromosome retained a wild-type p53 allele (Fig. 3a). Like Rb/p53-deleted lesions, Rb/p53-mutated PBs stained positive for synaptophysin but not GFAP (Fig. 3c, d), and in accordance with the R270H mutation, which stabilizes the protein, stained strongly for p53 (Fig. 3e).

To identify differences between these two closely related tumors, we performed GSEA[30] combined with Enrichment Map analysis[31,32]. Multiple immune response-related and metabolic pathways were enriched in Rb/p53-deleted relative to Rb/p53-mutated PBs, whereas cell cycle, mitotic replication, transcription, mRNA processing, and Bard1 (BRCA1-associated RING domain protein 1) pathways were enriched in Rb/p53-mutated compared with Rb/p53-deleted PBs (Supplementary Fig. 4a). In addition, four pathways involved in metastasis were enriched in the Rb/p53-mutated vs Rb/p53-deleted PBs (Fig. 3f and Supplementary Fig. 4b). In accordance, WAP-Cre:Rb[flox/flox]:p53[lsl_R270H/flox] mice exhibited not only metastases to the leptomeningeal surface of the spinal cord as seen in Rb/p53-deleted mice but also to leptomeningeal surfaces in the brain (Fig. 3g). Furthermore, Rb/p53-mutated PBs showed significantly more metastasis (60%) compared to Rb/p53-deleted (22%) (P < 0.0001; Fig. 3g), indicating

that the p53-R270H allele exerts a gain-of-function effect in this context.

**NOR predicted as therapeutic drug for PB.** Although new chemotherapy regimens have improved 5-year OS of PB patients, survival rates drop to 15% in ≤5-year-old patients, and tumors that shed metastases are virtually incurable[3]. There is therefore a pressing need to develop new approaches to treat metastatic PB. We used mRNA expression data from the two mouse PB models to calculate connectivity scores[33] using GSEA and genome-wide connectivity (GWC) mapping[34,35]. This in silico analysis consistently identified tricyclic drugs, some of which are FDA-approved antidepressants or antipsychotics, including NOR, promazine, norcyclobenzaprine and amitriptyline, as potential inhibitors for both Rb/p53-deleted and Rb/p53-mutated PBs (Fig. 4a; Supplementary Tables 2 and 3).

To investigate the effect of these antidepressant drugs, we established primary Rb/p53-deleted PB cells, cultured in serum-free media supplemented with EGF and FGF (Fig. 4b). Primary PB cells expressed synaptophysin, and sprouted secondary tumors following subcutaneous (sub. cu.) injection of $5 \times 10^5$ cells into immuno-deficient NOD/SCID mice (Fig. 4b). Secondary tumors were histologically similar to the primary PB with pleomorphic nuclei and scant cytoplasm, though rosettes were less obvious. Three independent primary PB cell lines (PB-1638, PB-1979, PB-2804) as well as human group 3 MB cell line, D425wt, showed significantly higher sensitivity to NOR, the top-ranked antidepressant drug, compared to immortalized HaCaT keratinocyte cells used as normal-like control (Fig. 4c).

NOR was shown to induce apoptosis in mouse bladder cancer cells at concentrations of ≥25 µM[36]. To determine whether NOR induced apoptosis in PB, cells were treated with NOR and tested for caspase-3/caspase-7 activation. NOR had no significant effect on caspase-3/7 activity after 1–2-h treatment, and only low induction of these caspases after 24 h (Fig. 4d, left; Supplementary Fig. 5a). In contrast, the anti-neoplastic cytotoxic drug gemcitabine, a nucleoside analog that interferes with DNA synthesis, induced robust caspase-3/7 activity (Fig. 4d, right, Supplementary Fig. 5a). Thus, short-term NOR treatment induces primarily a non-apoptotic cell death in PB cells.

To ask whether NOR mono-therapy could attenuate PB in vivo, we used MRI to monitor tumor growth. 70–80 day old WAP-Cre:Rb[flox/flox]:p53[flox/flox] mice were treated with vehicle alone or NOR (20 mg/kg, with gradual increase to 30 mg/kg and then 40 mg/kg, 5 days/week) for 5 weeks. MRI was performed just prior to and at the end of drug treatment, and tumor volume was computed by analyzing serial images (Supplementary Fig. 5b). On

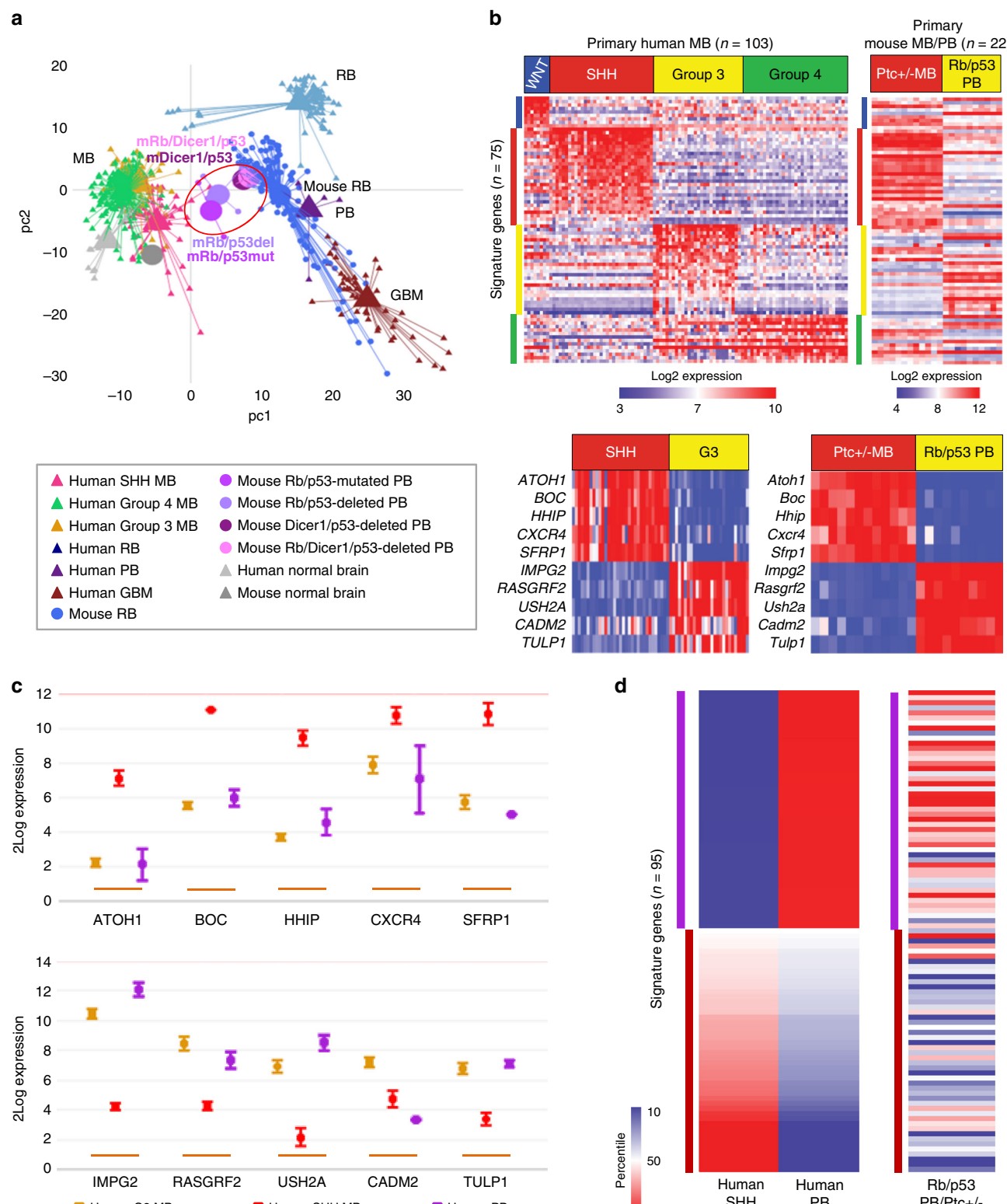

**Fig. 2 Mouse pineoblastomas cluster close to human pineoblastoma and medulloblastoma. a** GSEA-PC analysis of mouse Rb/p53-deleted (n = 12), Rb/p53-mutated (n = 5), Dicer1/p53-deleted (n = 3) and Rb/Dicer1/p53-deleted (n = 3) pineoblastomas (circled), showing close clustering with mouse retinoblastoma (mouse RB), human PB and medulloblastoma (MB) but not glioblastoma (GBM). A 3-D image is shown in Supplementary Fig. 3. **b** Top, 75 signature genes (depicted in supplementary Table S1) were used to classify human medulloblastoma into 4 subgroups (left) and compare mouse Rb/p53-deleted PBs with Ptc+/− mouse model of SHH MBs (right). Bottom, expression of top 10 signature genes from human SHH vs group 3 MB subgroups (left), and from mouse Ptc+/− MB vs Rb/p53-deleted PBs (right), showing similar relative expression. **c** Expression of top 10 signature genes (mean ± SD) identified above in human PBs, group 3 (n = 7) and SHH (n = 4) MBs in an independent dataset. **d** 95 signature genes (listed in supplementary Table S1) used to differentiate between human SHH MB and human PB. Similar relative expression of the same 95 signature genes in Rb/p53-deleted PBs, normalized to expression levels in the Ptc+/− mouse model of SHH MBs.

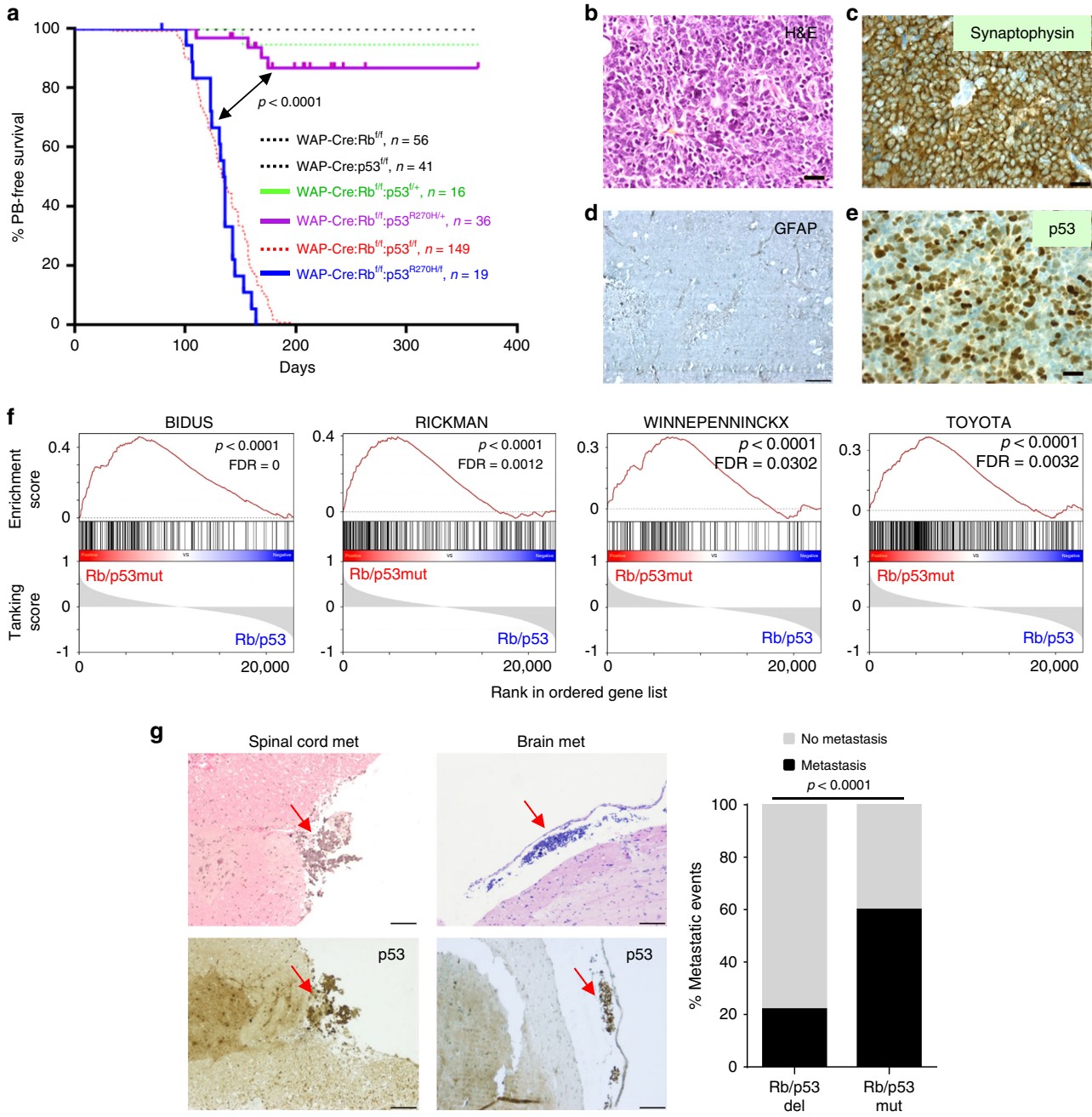

**Fig. 3 Rb deletion plus p53-R270H mutation induces highly metastatic pineoblastoma. a** Kaplan–Meier PB-free survival curves showing that WAP-Cre:Rb$^{flox/flox}$:p53$^{lsl\_R270H/flox}$ mice ($n = 19$, blue) developed PB with 100% penetrance and a median latency of 135.5 days, similar to WAP-Cre:Rb$^{flox/flox}$:p53$^{flox/flox}$ mice (shown in light dotted red line for comparison as in Fig. 1a). WAP-Cre:Rb$^{flox/flox}$:p53$^{lsl\_R270H/+}$ mice ($n = 36$, purple) also developed PB but with diminished penetrance (11.1%). Statistical analysis by Mantel-Cox test. **b** Representative H&E image of PB from WAP-Cre:Rb$^{flox/flox}$:p53$^{lsl\_R270H/flox}$ mice, showing similar histology to Rb/p53-deleted PBs. Scale bar, 2 μm. **c–e**. Representative IHC images of PB from WAP-Cre:Rb$^{flox/flox}$:p53$^{lsl\_R270H/flox}$ mice, showing strong expression of synaptophysin and p53, but no GFAP expression. Scale bar, 2 μm. **f** Gene set enrichment analysis (GSEA) revealing enrichment in pathways involved in metastasis in Rb/p53-mutated PBs compared to Rb/p53-deleted PBs. Full GSEA is provided in Supplementary Fig. 4. **g** Left, H&E (top) and p53 IHC (bottom) showing spinal cord and brain metastases in WAP-Cre:Rb$^{flox/flox}$:p53$^{lsl\_R270H/flox}$ mice. Scale bars, 2 μm. Right, frequency of metastatic events determined by H&E staining of multiple sections. Metastases were found in 60% ($n = 15$) of WAP-Cre:Rb$^{flox/flox}$:p53$^{lsl\_R270H/flox}$ mice compared to 21% ($n = 32$) in WAP-Cre:Rb$^{flox/flox}$:p53$^{flox/flox}$ mice. Scale bar, 2 μm. Chi-square analysis, $P < 0.0001$. For raw data see Source Data files.

average, NOR treatment suppressed tumor growth 7.03-fold compared to control mice (Fig. 4e, f).

**NOR acts as a CAD to inhibit autophagy.** Tricyclic antidepressants have been shown to eliminate breast cancer cells by interfering with serotonin transport[37], and GBM cells by promoting autophagy[38]. While tricyclic antidepressants inhibit serotonin and norepinephrine re-uptake transporters and act as antidepressants, they can also function as cationic amphiphilic drugs (CADs) that interfere with lysosomal function[39,40]. To determine the mechanism by which NOR exerts its inhibitory effect in PB, we first determined whether it acts as serotonin-norepinephrine reuptake inhibitor

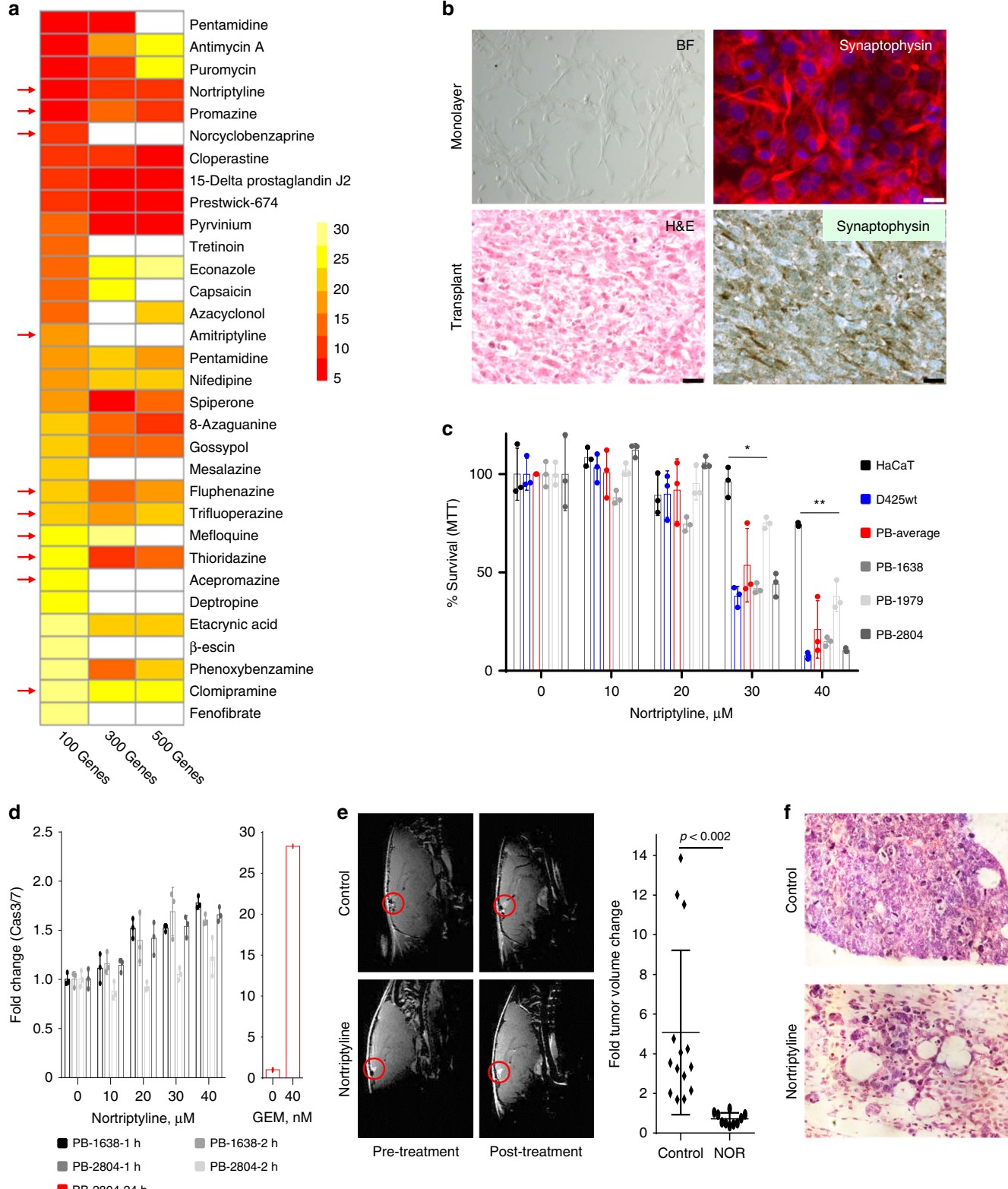

(SNRI) or as CAD. To this end, we analyzed NOR side-by-side with venlafaxine, an antidepressant drug that acts as SNRI but does not have CAD properties, and with astemizole, an anti-histamine drug that has CAD structure but no SNRI activity. Like NOR, astemizole (CAD) effectively suppressed mouse PB cell growth whereas venlafaxine (SNRI) did not (Fig. 5a), suggesting that NOR exerts its effect through its cationic amphiphilic activity.

We next sought to determine whether NOR promoted autophagy as was previously suggested for the effect of imipramine on gliomas[38]. Imipramine, which ranked at the top 100–300 of 1288 drugs, depending on the parameters in our connectivity map analysis, suppressed growth of primary PB cells, albeit much less effectively than NOR (Supplementary Fig. 6a). As markers for autophagy, we used two factors that execute autophagy and are

**Fig. 4 Nortriptyline predicted as a potential therapeutic for pineoblastoma. a** Connectivity mapping by GWC of mouse Rb/p53-deleted PB identifies multiple tricyclic drugs (red arrows) with nortriptyline (NOR) topping the list. These drugs including NOR ranked high in similar analysis with Rb/p53-mutated PB, and using connectivity mapping by GSEA (Supplementary Tables 2 and 3). Color scale bar represents top 30 drugs when analyzing 100 genes per signature. **b** Bright field (BF) image and synaptophysin expression in primary Rb/p53-deleted PB cells cultured as monolayer cells. Original magnification, ×400. Flank tumors, developed following sub. cu. transplantation of primary Rb/p53-deleted PB into NOD/SCID mice, exhibit similar histology (H&E staining) and synaptophysin expression as primary lesions. Scale bars, 2 μm. **c** Response of human HaCaT cells, Group 3 medulloblastoma D425wt cells and three primary mouse Rb/p53-deleted PB cells (PB-1638, PB-1979, and PB-2804) to increasing concentrations of NOR, 24-h post treatment using MTT assays. $n = 3$ for each group. One-way ANOVA, *$P < 0.05$, **$P < 0.01$. **d** Left, marginal caspase3/7 activity in mouse Rb/p53 PB-1638 and PB-2804 cells in response to increasing concentrations of NOR treatment for 1–2 h. Right, gemcitabine, used as a positive control, induces robust caspase3/7 activity. $n = 3$ for PB-1638-1 h, PB-1638-2 h, PB-2804-1 h, PB-2804-2 h. Low caspase3/7 activity after 24-h NOR treatment is shown in Supplementary Fig. 5a. **e** Left, representative MRIs of WAP-Cre:Rb^flox/flox^:p53^flox/flox^ mice treated with vehicle control ($n = 14$) or NOR ($n = 10$; 20 mg/kg/day, i.p. with gradual increase to 40 mg/kg/day), 5 days/week for 5 weeks. Right, fold tumor volume change compared to initial tumor volume. Two-tailed unpaired student's t-test, $P < 0.002$. PBs are indicated by red circles. **f** Histology of residual PB following treatment with NOR vs vehicle control in vivo. Original magnification, ×400. Results in (**c**), (**d**) and (**e**) are representative of three independent experiments. Results are presented as mean ± SD in (**c**), (**d**), and (**e**). For raw data see Source Data files.

themselves degraded by this process: the ubiquitin-binding protein p62-sequestosome 1/SQSTM1, which targets proteins for selective autophagy; and microtubule-associated proteins 1 A/1B light chain 3B/LC3B, involved in autophagosome biogenesis[41]. Western blot analysis revealed accumulation of both proteins following 24-h NOR treatment (Fig. 5b; Supplementary Fig. 6b). p62 level remained high 48-h post-treatment (Supplementary Fig. 6b), suggesting autophagy blockade. Immunofluorescent staining demonstrated strong and sustained induction and colocalization of p62 with LC3B (Fig. 5c; Supplementary Fig. 6c). Lamp-1 (lysosome-associated membrane protein 1), a lysosome membrane specific marker, did not show robust induction following NOR treatment but was colocalized with LC3b and p62 (Fig. 5c), suggesting increased level of autolysosome formation.

To further investigate the effect of NOR on autophagic flux, we used the tandem fluorescent-tagged LC3 reporter construct, mRFP-GFP-LC3, and the differential stability of GFP and mRFP fluorescent proteins at different pH[42]. During autophagy, autophagosomes with their engulfed cargo fuse to lysosomes to form autolysosomes where their cargo is degraded[43,44]. The lysosomes and autolysosomes are highly acidic providing optimal conditions for lysosomal hydrolases such as cathepsin proteases. Whereas mRFP-LC3 is stable both in autophagosomes and autolysosomes, GFP-LC3 is inactive under acidic conditions and thus fluoresces only in autophagosomes. Thus, in cells expressing mRFP-GFP-LC3, autophagosomes are labeled yellow (both GFP and mRFP), whereas autolysosomes are red (mRFP). Primary Rb/p53-deleted PB cells were transfected with mRFP-GFP-LC3 and treated with the autophagy inhibitor chloroquine (CQ), autophagy inducer rapamycin (RAP), vehicle control or NOR for 2 or 6 h. We observed increased percentage of yellow puncta in NOR treated cells for both time points (Fig. 5d; Supplementary Fig. 7), indicating blockade of autophagic flux at the autophagosome stage or, as shown below, at the autolysosome stage due to reduced acidity and impaired function.

CQ inhibits autophagy by deacidifying the lysosomes and inhibiting its fusion to autophagosomes[45]. To determine whether NOR can synergize with CQ, Rb/p53-deleted PB cells were treated with vehicle alone, low CQ concentration (20 μM), or increasing doses of NOR either alone or together with CQ for 24 h followed by MTT assays to quantify cell viability. Synergy was calculated using CompuSyn analysis [http://www.combosyn.com][46], with combination index (CI) of <0.85 denoting synergy, CI = 0.9–1.0 additive, and CI > 1.1 antagonistic effects. Within the linear range of inhibition by NOR alone, NOR plus CQ showed strong synergy in suppressing growth of Rb/p53-deleted PB cells, suggesting that these drugs suppress autophagy and/or cell survival by different, yet complementary mechanisms (Fig. 5e; Supplementary Fig. 6d).

**NOR disrupts the lysosome, impeding autolysosome function.** The aforementioned results suggested that rather than inducing autophagy, NOR suppresses this process by acting as a CAD. CADs were previously shown to increase lysosomal membrane permeability (LMP)[47], and could thereby, like CQ, inhibit the autophagic flux by disrupting the lysosome. To test for such an effect in PB, we monitored the rate of lysosomal release of acridine orange in response to NOR. Exogenously added acridine orange enters lysosomes where it gets protonated, leading to a shift in its emission spectrum from green to red; it further sensitizes lysosomal membrane to photo-oxidation by blue light[39]. When LMP is induced by photo-oxidation, acridine orange leaks into the cytosol, causing loss of punctate red signal and increase in diffused cytoplasmic green signal. To determine whether NOR increases LMP, Rb/p53-deleted PB cells were treated with NOR or vehicle control for 2 h, acridine orange dye was added and the cells were subjected to photo-oxidation. Time-lapse microscopy revealed a significant increase in the rate of green fluorescent intensity, i.e., acridine orange release from lysosomes, in NOR treated cells compared to control (Fig. 6a), indicating LMP induction by NOR.

Lysosomal pH dictates the position of lysosomes within the cell; peripheral lysosomes are less acidic than juxtanuclear lysosomes[48]. LMP reduces acidity of lysosomes and shifts them to the periphery. To determine the effect of NOR on lysosome localization, Rb/p53-deleted PB cells were treated with NOR for 2 h, acridine orange was added and the cellular localization of lysosomes was examined under confocal microscopy. NOR treatment significantly increased the fraction of peripheral lysosomes from 41 to 60% (Fig. 6b). LMP also affects lysosomal volume and vice versa[49]. To determined whether NOR affects lysosomal size, PB cells were treated with increasing concentrations of NOR for 2 h and localization of cathepsin B and Lamp-1 was assessed by immunofluorescent (IF) staining. Lysosomal volume increased dramatically in proportion to NOR concentration (Fig. 6c). To track lysosomes and cathepsin B activity in live cells, we employed LysoTracker Green (LTG) and Magic Red Cathepsin B, respectively. Increased NOR concentration diminished the punctate green signal intensity of lysosomes, and increased cathepsin B activity in the cytosol, which stained yellow, indicating overlap of the two fluorescent dyes (Fig. 6d). In addition, cell fractionation analysis demonstrated NOR treatment increased levels of cathepsin B in the cytosol, while total cellular level of this protease remained unchanged (Fig. 6e; Supplementary Fig. 8a).

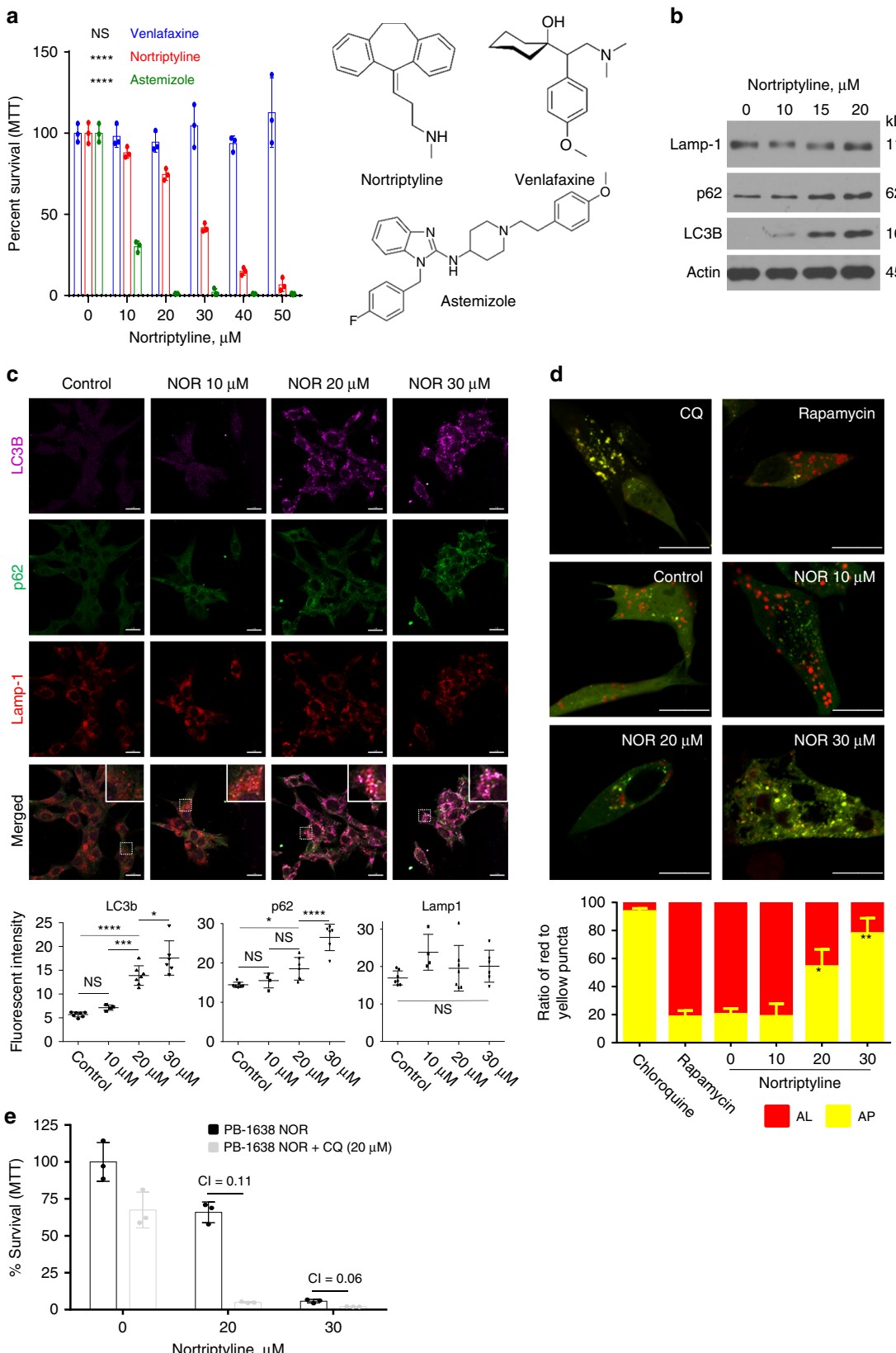

Finally, TEM revealed NOR induced accumulation of large autophagic vacuoles containing partially degraded contents, lipid droplets and laminar bodies, indicative of defective autolysosomes (Fig. 6f; Supplementary Fig. 8b). Taken together, these results suggest that NOR induces LMP, leading to proton and cathepsin B release, and accumulation of enlarged, non-functional autolysosomes, leading primarily to non-apoptotic cell death.

**NOR synergizes with gemcitabine to suppress PB.** Given the similarity between PB and group 3 MB, and the sensitivity of

**Fig. 5 Nortriptyline impairs autophagy in pineoblastoma cells. a** MTT assays of mouse Rb/p53 deficient PB cells treated for 24 h with vehicle control or increasing concentrations of venlafaxine, nortriptyline (NOR) or astemizole (chemical structures on the right). $n = 3$ for each group. One-way ANOVA, ****$P < 0.0001$, n.s. = not significant. **b** Immunoblots of mouse PB cells treated with vehicle control or increasing concentrations of NOR for 24 h, showing elevated p62 and LC3B but not Lamp-1 expression. **c** Top, mouse PB cells treated with vehicle control or increasing concentrations of NOR for 2 h and then immunostained for LC3B (purple), p62 (green), and Lamp-1 (red). Bottom, fluorescent intensity quantified using ImageJ. Scale bar, 20 μm. One-way ANOVA, *$P < 0.05$, ***$P < 0.001$, ****$P < 0.0001$, n.s. = not significant. **d** Top, assessment of autophagosome (AP, yellow) and autolysosome (AL, red) puncta using mRFP-GFP-LC3 reporter plasmid transfected into mouse PB cells, followed by treatment with chloroquine (CQ, 20 μM), rapamycin (500 nM), vehicle control or increasing concentrations of NOR for 2 h. Bottom, intensities of yellow and red puncta measured using ImageJ with JACoP plug-in. Scale bar, 20 μm. Two-tailed student's $t$-test, *$P < 0.05$, **$P < 0.01$. Additional cell lines and duration of treatments are shown in supplementary Fig. S7. **e** MTT assays of mouse PB cells treated with vehicle control ($n = 3$), chloroquine ($n = 3$; 20 μM), increasing concentrations of NOR alone ($n = 3$), or together with chloroquine ($n = 3$; 20 μM). Synergism was calculated using CompuSyn (CI < 0.85 denotes synergy). Results are presented as mean ± SD in (**a**), (**c**), (**d**), and (**e**). See Source Data files for raw data.

the latter to pemetrexed and gemcitabine (GEM)[50], we analyzed the effects of these antineoplastic drugs on Rb/p53-deficient PBs. We also analyzed inhibitors of Skp2 (SMIP004 and Skp2 III), which effectively prevents spontaneous tumorigenesis in Rb$^{+/-}$ mice[51]; cisplatin, currently used to treat PB, GBM, PNETs and ependymomas; doxorubicin; and the PI3K/mTOR inhibitor BEZ235. Gemcitabine had the lowest $IC_{50}$ value (31.3 nM; Fig. 7a), and was equally effective against Rb/p53-deleted and Rb/p53-mutated PBs in vitro (Supplementary Fig. 9a). To determine whether gemcitabine is effective against PB in vivo, 70–80-day-old WAP-Cre:Rb$^{flox/flox}$:p53$^{flox/flox}$ mice were treated with 60 mg/kg GEM (i.v. injection, once a week for 5 weeks) and MRI was performed on all mice at starting and end points. GEM significantly suppressed PB growth 5.18-folds over this period (Fig. 7b).

Inhibition of autophagy potentiates the effect of anti-neoplastic cytotoxic drugs[52,53]. Indeed, NOR strongly synergized with GEM in eliminating Rb/p53-deleted PB cells as well as human group 3 MB cells in vitro (Fig. 7c; Supplementary Fig. 9b). Annexin V/PI flow cytometric analysis revealed that NOR plus GEM cooperated to induce necrotic or late apoptotic, Annexin V$^+$/PI$^+$, cell death after 24 h of treatment (Supplementary Fig. 9c). To investigate the effect of NOR plus GEM combination therapy on PB in vivo, weaned, 30 day old WAP-Cre:Rb$^{flox/flox}$:p53$^{flox/flox}$ mice were injected with vehicle alone, NOR (20 mg/kg/d, i.p.), GEM (60 mg/kg/week, i.v.) or both drugs for 40 days. MRI analysis revealed a significant inhibition in all treatment arms compared with sham control (Fig. 7d). NOR plus GEM treatments had the most significant effect, reducing tumor volume 4.26-fold compared to control ($P = 0.0004$).

To corroborate these results with longer and continuous follow-up, primary Rb/p53-deleted PB cells were subcutaneously injected into NOD/SCID mice and, when tumor became palpable, mice were treated with vehicle, NOR (10 mg/kg/d, i.p.), GEM (60 mg/kg/week, i.v.) or NOR plus GEM. To assess for toxicity, mouse weight was monitored throughout the course of treatment. Although there was a trend toward reduced body weight, none of these treatments had deleterious side effects (Supplementary Fig. 9d). NOR plus GEM treatment significantly reduced tumor growth (Supplementary Fig. 9d), and extended median survival 3.29-fold relative to control, 2.14-fold relative to GEM, and 2.14-fold relative to NOR (Fig. 7e). Thus, NOR and GEM effectively cooperate to suppress PB growth in preclinical models.

**Modeling *DICER1*-deficient PB**. In addition to *RB1* loss, germline mutation in *DICER1*, a ribonuclease involved in microRNA processing and DNA repair, also predisposes children to PB[6–8]. Using WAP-Cre, we deleted a floxed allele of Dicer1, flanking the second RNasIII domain[54], either alone or together with p53. Six of nineteen WAP-Cre:Dicer1$^{flox/flox}$:p53$^{flox/flox}$ mice (31.6%) developed PB with incomplete penetrance by 270 days of age as

endpoint (Fig. 8a). For those mice that developed PB, the median latency was 151.5 days. WAP-Cre:Dicer1$^{flox/+}$:p53$^{flox/flox}$ mice ($n = 16$) did not develop such lesions, indicating that both alleles of Dicer1 must be disrupted to promote PB. WAP-Cre:Dicer1$^{flox/flox}$ and WAP-Cre:Dicer1$^{flox/+}$ mice with wildtye p53 also did not develop PB.

Although the latency of WAP-Cre:Dicer1$^{flox/flox}$:p53$^{flox/flox}$ mice was significantly longer than that of WAP-Cre:Rb$^{flox/flox}$:p53$^{flox/flox}$ mice ($P < 0.0001$), they developed full blown PB with characteristic pleomorphic nuclei, scarce cytoplasm and rosette structures (Fig. 8b). Despite their substantial size, most Dicer1/p53-deficient PBs did not protrude the brain as did Rb/p53-deleted lesions.

We also generated WAP-Cre:Dicer1$^{flox/flox}$:Rb$^{flox/flox}$:p53$^{flox/flox}$ mutant mice with deletion of all three tumor suppressors. These mice developed bulging PB that initiated earlier than Rb/p53-deficient tumors but showed reduced penetrance (13 of 14 mice; 92.9%) by 270 days of age, and wider latency (median latency 138 days), yet with similar histology as Rb/p53 and Dicer1/p53 lesions (Fig. 8a–b). Deletion of Dicer1 and p53 but not Rb in WAP-Cre:Dicer1$^{flox/flox}$:p53$^{flox/flox}$ mice, and deletion of all three genes in WAP-Cre:Dicer1$^{flox/flox}$:Rb$^{flox/flox}$:p53$^{flox/flox}$ mice were confirmed by PCR (Fig. 8c).

To molecularly assess Rb pathway loss in these PBs, we performed pathway activity analysis using an RB knockout signature[55]. As expected, Rb/p53- and Rb/Dicer1/p53-deletion PBs exhibited high RB pathway loss compared to control brain tissue (Fig. 8d). In contrast, only one Dicer1/p53 PB exhibited similar increase in RB loss signaling, whereas the other two tumors showed elevated but not robust RB loss activity. E2F1-3 pathway activation analysis[55] revealed elevated signaling in one or more of these transcription factors downstream of RB-loss in each Dicer1/p53 lesion. There was very high activity of E2F2 in the single PB with robust RB loss signaling, which may reflect an oncogenic induction in this particular tumor.

GSEA-PC analysis revealed that Dicer1/p53 and Rb/Dicer1/p53 PBs clustered closely with Rb/p53-deficient PBs and mouse RB as well as with human PB, but away from GMB (Fig. 2a, Supplementary Fig. 3). Thus, the WAP-Cre transgene can be used to model germline mutations in PB, and combined deletion of Rb-p53 o rDicer1-p53 via this deleter line induces histologically similar PBs with distinct kinetics as observed in children[10].

**General sensitivity of PB to NOR**. To identify potential therapeutic targets for Dicer1-defcient PB, we performed CMAP analysis on the Dicer1-mutant PBs as described in Fig. 4a for Rb/p53-deficient lesions. NOR and other tricyclic/anti-psychotic medicine (chlorpromazine, clomipramine, fluphenazine and norcyclobenzaprine) appeared again among the top predicted drugs for both Dicer1/p53 and, to a lesser extent, Rb/Dicer1/p53

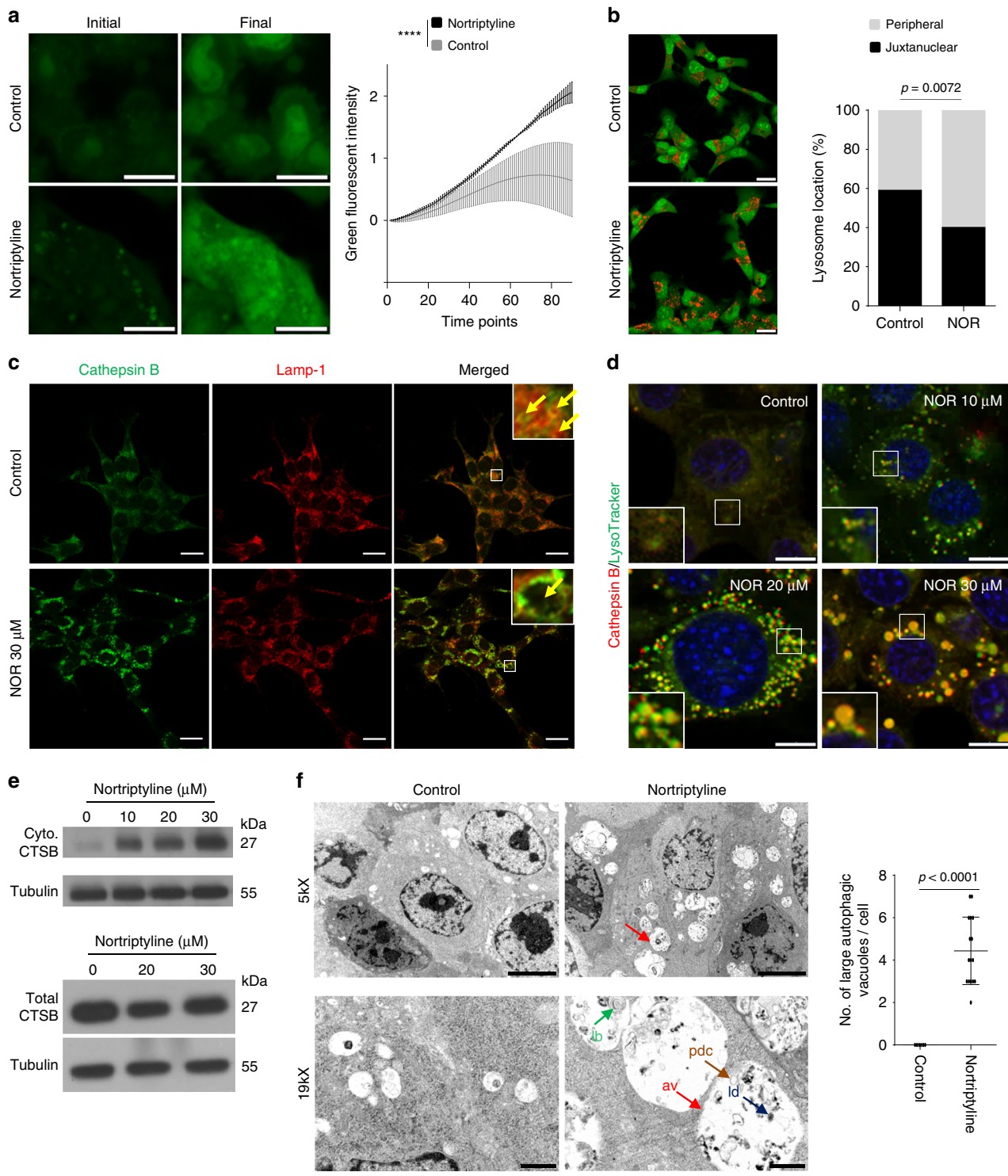

PBs (Fig. 8e; Supplementary Tables 4 and 5). The Oxidative Phosphorylation (OXPHOS) inhibitor, antimycin A, was also predicted to suppress PBs from all three models. Rb/Dicer1/p53-deficient, but not Dicer1/p53-deficient, primary PB cells could readily be cultured in vitro and they exhibited high sensitive to NOR (Fig. 8f).

Finally, very few human PB cell lines have been successfully established in culture. One line, TS13-19, isolated from sporadic PB and propagated as non-adherent tumorspheres in serum-free media[56], was also highly sensitive to NOR (Fig. 8g). Together,

these results demonstrate that mouse and human PBs with different oncogenic drivers exhibit elevated sensitivity to NOR/lysosome disruption, indicating the potential benefit of this drug for diverse types of PB.

## Discussion

We report the development of tractable immune-competent mouse models for the two major germline mutations that drive PB in children, *RB1* and *DICER1*. We employed a WAP-Cre deleter line, commonly used to target mammary lobular-alveola

**Fig. 6 Nortriptyline disrupts the lysosome in pineoblastoma cells. a** Acridine orange release assay in mouse PB cells treated with vehicle control ($n = 3$) or 30 µM NOR for 2 h. Changes in green fluorescent intensity at different time points measured using ZEISS LSM 700 confocal microscope. Scale bar, 10 µm. Two-tailed paired student's $t$-test, ****$P < 0.0001$. **b** Lysosome localization assay. Mouse PB cells were treated with vehicle control or NOR (12.5 µM for 2 h). Acridine orange was added and location of lysosomes relative to nuclei determined. Scale bar, 20 µm. Chi-Square test, $P = 0.0072$. **c** Representative IF analysis for lysosomal membrane permeability using cathepsin B (green) and Lamp-1 (red) antibodies in PB cells treated with vehicle control or 30 µM NOR. Inlets, normal (top) vs enlarged (bottom) images of lysosomes (yellow arrows). Scale bar, 20 µm. **d** Lysosome integrity and cathepsin B activity assays. PB cells treated with vehicle control or increasing concentrations of NOR for 2 h were analyzed by LysoTracker and Cathepsin B Magic Red Kit under ZEISS LSM700 confocal microscope. Scale bar, 10 µm. **e** Top, Western blot analysis of cathepsin B expression in cytoplasmic fractions (Cyto.) in PB 2804 cells treated with increasing concentration of NOR for 2 h. Bottom, total cathepsin B levels in whole cell lysates remained relatively unchanged. **f** Representative TEM images of PB cells treated with vehicle control ($n = 6$) or 30 µM NOR ($n = 16$) for 2 h showing increased number of autophagic vacuoles compared to control cells (top left and top right). Autophagic vacuoles (red arrow) are filled with lipid droplets (blue arrow), partially degraded contents (brown arrow) and lamellar bodies (green arrow; bottom right). Scale bars, 5 µm (top panel) and 1 µm (bottom panel). Bar graph comparing the number of large autophagic vacuoles per cell by TEM, using 6 control and 16 NOR treated cells. Two-tailed $t$-test, $P < 0.0001$. Results are presented as mean ± SD in (**a**) and (**f**). See Source Data files for raw data.

during estrous cycle and pregnancy[13]. The initial publication on this transgene demonstrated low expression in whole brain tissue. Our results now demonstrate that this transgene is expressed in the developing pineal gland and, to a lesser extent, in the pituitary gland. This is likely due to erroneous expression of this transgene at the specific integration site in this line as we observed only marginal expression of endogenous WAP gene in the pineal gland. PBs were observed following WAP-Cre-mediated deletion of Rb plus p53 or Dicer1 plus p53, but not after deletion of any of these genes alone or following combined disruption of Pten plus p53, indicating a unique sensitivity of pineal gland cells to combined disruption of Rb or Dicer1 plus p53.

Nonetheless, various WAP-Cre-based transgenic mice may only induce benign lesions or have subtle effects on the pineal gland. This gland plays a central role in coordinating circadian rhythms by releasing melatonin. In turn, melatonin modulates renewal of hematopoietic stem and progenitor cells[57], controls the circadian clock, and its perturbation has been implicated in different types of cancer, including those of the breast[58]. These considerations should be taken into account when using WAP-Cre to study tumorigenesis of the mammary epithelium.

Consistent with the short median latency of RB1 versus DICER1 PB subtypes in children[10], WAP-Cre-mediated deletion of Rb plus p53 had a shorter latency and 100% penetrance compared with longer latency and incomplete penetrance of the Dicer1/p53 model. The Dicer1 mouse used here deletes most of the second RNasIII domain upon Cre-mediated deletion, yet the deletion is in-frame, yielding a defective but stable protein[54]. miRNA-independent roles of Dicer1 have been demonstrated suggesting the truncated, RNasIII deleted, Dicer1 protein may still be functional in microRNA-independent contexts that may impact tumorigenesis[59]. It would therefore be important to determine whether complete homozygous deletion of Dicer1 would increase the penetrance of PB.

To identify the cell of origin of these PBs, we have performed lineage-specific marker analysis. In preliminary results, micro-tumors from 30 day old Rb/p53-deleted mice as well as full-blown tumors stained positive for 5-hydroxytryptamine receptor (5-HT, serotonin receptor), which marks matured pinealocytes[60], but were completely negative for pax6 (pinealocyte precursor cells), nestin (neuronal stem cells) and the microglia marker ox42. Thus, combined deletion of Rb/p53 or Dicer1/p53 in pinealocytes may induce partial dedifferentiation leading to reduced expression of 5-HT. Assignment of pinealocytes as the cells of origin is also consistent with the ability of IRBP transgenic mice to induce PB, albeit at low frequency, following Rb deletion or following over-expression of cyclin D1, the latter of which induces pRb phos-phorylation and inactivation, on p53 null background[12,29].

Notably however, Cyclin D1 has other targets in addition to cell cycle control[61], and there is no evidence so far that this cyclin or other D type cyclins are amplified in PB. It is also possible that combined loss of Rb and p53 induces partial trans-differentiation of another lineage into weakly 5-HT-expressing cells. Similar considerations were made during the search for a cell of origin for RB. Definite assignment involved systemic functional analysis of various retinal precursors, showing that RB1 knockdown in post-mitotic human cone precursors but not other progenitors induced cell proliferation, leading to tumors with features of RB following orthotopic transplantation[62].

Our in silico drug prediction analysis identified multiple tri-cyclic, antidepressant drugs such as NOR, which ranked at the top, as potential therapeutics for Rb/p53-deleted and Rb/p53-mutated PBs as well as for Dicer/p53- and Rb/Dicer1/p53-defi-cient lesions. NOR inhibited growth of primary Rb/p53- and Rb/Dicer1/p53-deficient PB cells as well as a human PB cell line[56]. NOR suppressed autophagy flux not by blocking assembly of the autolysosome but by disrupting the lysosome. Lysosome disrup-tion caused cathepsin release and reduced acidity, leading to accumulation of large, non-functional autolysosomes and largely non-apoptotic cell death. NOR induced accumulation of lyso-somes, cathepsin B activity and expression of the pro-autophagy factors LC3B and p62, likely as a result of an auto-regulatory feedback response to defective lysosomes and reduced autophagic flux (Fig. 9). Lysosome disruption may be superior to autophagy inhibitors as it induces additional lethal effects such as cathepsin-release[63].

NOR is given as anti-depressant at maximal dose of 150 mg/day, which is, for an average 75 kg person = ~2 mg/kg. Small animals metabolize drugs faster than large animals, and mouse doses can be as high as 12.3 times more than humans[64]. This translates into NOR dose of 2 mg/kg × 12.3 = 24.6 mg/kg for mice. For NOR treatment alone, we started with 20 mg/kg and slowly increased the dose to 40 mg/kg, whereas in combination with GEM, we kept the dose at 20 mg/kg (mouse models) or 10 mg/kg (flank assays). Thus, NOR doses used here for mice are directly applicable to humans. Of note, meta-analysis of tricyclic antidepressant users found a significant reduction in incidence of glioma and colorectal cancer[65]. Moreover, for lung cancer, tri-cyclic antidepressants or antihistamines significantly improved survival[40,66].

In conclusion, we developed new preclinical models for the most prevalent germline mutations that drive PB; RB1 and DICER1, as well as for combined RB1 plus DICER1 loss on p53 null or mutant background. These immune-competent models should be instrumental in evaluating mechanism of dissemination, identifying cell of origin and candidate new therapeutics targeting

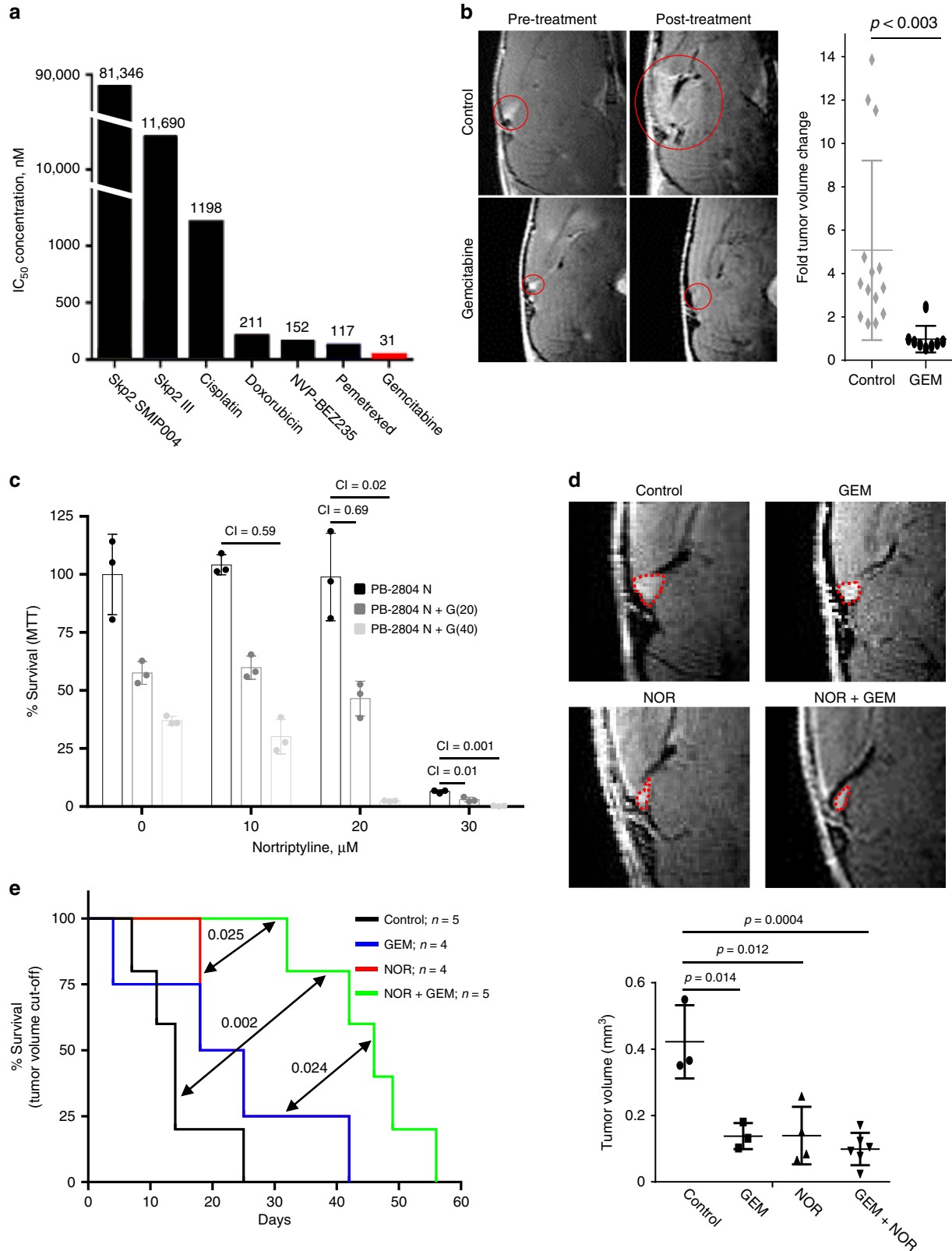

the immune system and both primary and metastases. We showed that diverse mouse and human PB cells are highly sensitivity to the FDA-approved tricyclic antidepressant NOR, which disrupts the lysosome, leading to inhibition of autolysosome function, cathepsin release, and cell death. Lastly, we demonstrated that NOR synergizes with gemcitabine to effectively suppress PBs in our preclinical model, suggesting a new therapeutic approach for this rare, yet lethal pediatric cancer.

## Methods

**Generation of Rb-deficient and Dicer1-deficient mouse model.** All mouse experiments were performed in accordance with the current Canadian Animal

**Fig. 7 Nortriptyline synergises with gemcitabine to suppress pineoblastoma. a** $IC_{50}$ values (in nM) following MTT assays of mouse Rb/p53 PB cells treated with indicated drugs. Gemcitabine was most potent ($IC_{50} = 31$ nM). **b** Left, representative MRIs of WAP-Cre:Rb$^{flox/flox}$:p53$^{flox/flox}$ mice treated with vehicle control ($n = 14$) or gemcitabine ($n = 8$; 60 mg/kg/week, i.v.) for 5 weeks. Right, fold tumor volume change compared to initial tumor volume. These experiments were performed side-by side with those described in Fig. 4e, and the control arm is the same. Two-tailed unpaired Student's $t$-test, $P < 0.003$. PBs are indicated by red circles. **c** MTT assays of mouse PB cells treated with vehicle control ($n = 3$), gemcitabine alone ($n = 3$), increasing concentrations of NOR alone ($n = 3$) or co-treatment with gemcitabine ($n = 3$; 20 nM or 40 nM). Cooperation Index was calculated using CompuSyn (CI < 0.85 indicates synergy). **d** Representative MRIs of WAP-Cre:Rb$^{flox/flox}$:p53$^{flox/flox}$ mice treated with vehicle control ($n = 3$), gemcitabine ($n = 3$; 60 mg/kg/week, i.v.), NOR ($n = 4$; 20 mg/kg/day, i.p.), or both drugs ($n = 6$). Treatments commenced at 30 days of age and MRIs were taken 40 days later. Statistical analysis by two-tailed unpaired student's $t$-test. PBs are indicated by red dashed circles. **e** Kaplan–Meier survival curve of NOD/SCID mice following sub. cu. injections of primary PB cells, treated with control, gemcitabine (60 mg/kg/week, i.v.), NOR (10 mg/kg/d, i.p.) or both drugs. Endpoint was set as tumor volume > 550 mm$^3$. Statistical analysis by Mantel-Cox test. Results in (**c**) are representative of three independent experiments. Results are presented as mean ± SD in (**b**), (**c**), and (**d**). See Source Data files for raw data.

Care Council guide for the care and use of laboratory animals and were approved by the Toronto General Research Institute Animal Research Committee, UHN. To generate experimental and control mice, WAP-Cre:Rb$^{flox/flox}$ mice and p53$^{flox/flox}$ or p53$^{LSLR270H/+}$ mice were bred to obtain WAP-Cre:Rb$^{flox/+}$:p53$^{flox/+}$ and WAP-Cre:Rb$^{flox/+}$:p53$^{LSLR270H/+}$ mice, respectively. Subsequently, WAP-Cre:Rb$^{flox/+}$:p53$^{flox/+}$ mice were interbred to generate WAP-Cre:Rb$^{flox/flox}$ mice, WAP-Cre:p53$^{flox/flox}$ mice and WAP-Cre:Rb$^{flox/flox}$:p53$^{flox/flox}$ mice. WAP-Cre:Rb$^{flox/+}$:p53$^{LSLR270H/+}$ mice and WAP-Cre:Rb$^{flox/+}$:p53$^{flox/+}$ mice were bred to generate WAP-Cre:Rb$^{flox/flox}$:p53$^{LSLR270H/+}$ and WAP-Cre:Rb$^{flox/flox}$:p53$^{LSLR270H/f}$ mice. Control and experimental mice were on mixed genetic backgrounds. Mouse genotyping was performed as previously described[14]. The mT/mG, a double fluorescent Cre reporter mouse, generously provided by Dr. Hitoshi Okada, was crossed with WAP-Cre:Rb$^{flox/flox}$:p53$^{flox/flox}$ mice to generate WAP-Cre:Rb$^{flox/+}$:p53$^{flox/+}$: mT/mG mice. Subsequently, WAP-Cre:Rb$^{flox/+}$:p53$^{flox/+}$:mT/mG mice were interbred to generate WAP-Cre: Rb$^{flox/flox}$:p53$^{flox/flox}$:mT/mG mice. Dicer1-floxed mice (#006366) obtained from the Jackson Laboratory were bred to generate WAP-Cre:Dicer1$^{flox/flox}$, WAP-Cre:Dicer1$^{flox/flox}$:p53$^{flox/flox}$, WAP-Cre:Dicer1$^{flox/flox}$:p53$^{flox/flox}$:mT/mG and WAP-Cre:Dicer1$^{flox/flox}$:Rb$^{flox/flox}$:p53$^{flox/flox}$ mice.

**Cell culture and drug treatments**. Primary tumor cell lines were established from WAP-Cre:Rb$^{flox/flox}$:p53$^{flox/flox}$, WAP-Cre:Rb$^{flox/flox}$:p53$^{LSL\_R270H/flox}$ and WAP-Cre:Dicer1$^{flox/flox}$:Rb$^{flox/flox}$:p53$^{flox/flox}$ mice as described below. PB lesions were surgically removed, dissociated into single cells using accutase (Thermo Fisher Scientific, #A1110501), passed through 40 µm cell strainer (BioShop, #SP104151), and plated on laminin-coated Primaria plate (VWR, #47743-734; Fisher scientific, #08-772-4 G). D425wt is a human Group 3 MB cell line (MDT). TS13-19 is a human PB cell line[56], All cell lines were cultured in Neurobasal medium (Gibco, #LS21103049) supplemented with 2mM L-Glutamine (Gibco, #25030081), 1X B27 (Gibco, #154044), 10 ng/ml rhEGF (Gibco, #PHG0311L), 10 ng/ml bFGF (Gibco, #PHG0266), 75 µg/ml BSA (BioShop, #ALB001.50), 2 µg/ml heparin (BioShop, #HPA333.100) and 1X Penicillin-Streptomycin (Gibco, #15140122).

For MTT assay, the following drugs were used: NOR (Sigma, #N7261), astemizole (Sigma, A2861), venlafaxine (Selleckchem, #S1441), imipramine (Sigma, #I0899), chloroquine (BioShop, #CHL919), rapamycin (BioShop, #RAP004), Skp2 SMIP004 (EMD Millipore, #500517), Skp2 III (EMD Millipore, #506305), cisplatin (Selleckchem, #S1149), NVP-BEZ235 (Selleckchem, #S1009), pemetrexed (Selleckchem, #S1135), and gemcitabine (Selleckchem, #S1149).

$1 \times 10^4$ primary tumor cells were plated on 96-well Primaria plate coated with laminin. After 24 h, cells were treated with vehicle control or drugs of interest for 24 h. After treatment, MTT solution was added, incubated for 3 h, solutions were aspirated, 100 µl DMSO was added and $OD_{570}$ readings were taken with Vmax Kinetic Microplate Reader. Synergy was calculated using CompuSyn software, where combination index (CI) of <0.85 was considered as synergistic, CI = 0.9–1.0 as additive, and CI > 1.1 as antagonistic.

For confocal and two-photon microscopy, cells were plated on Nunc Lab-Tek II Chamber slide (Sigma, #C7057) coated with laminin.

**mRFP-GFP-LC3 reporter assay**. Primary tumor cells were transfected with mRFP-GFP-LC3 using TurboFect Transfection Reagent (ThermoFisher, #R0531). After transfection, cells were treated with chloroquine (20 µM), rapamycin (500 nM), vehicle control or increasing concentrations of NOR. Intensities of yellow and red puncta were measured using ImageJ with JACoP plug-in.

**Gross histological analysis**. Brains and spinal cords were dissected from normal and tumor-bearing mice carrying mT/mG reporter. Tissues were fixed in 10% formalin for >24 h. Gross anatomy and fluorescence signals were examined using the CRI Maestro system.

**Histopathological, IHC/IF, and immunoblot analysis**. Mouse brains were dissected along with skull, fixed in 10% formalin, decalcified in 0.5 M EDTA, and incubated in 30% sucrose solution. The brain and skull were cut into 250 µm

sections using Leica SM2000R sliding microtome, and sections were viewed under ZEISS LSM710 two-photon microscope to locate micro-lesions. For H&E staining, tissues were cut as 5 µm sections. The following primary antibodies were used to conduct IHC and IF staining: p53 (1:100, Santa Cruz, C, #SC-126), synaptophysin (1:200, Cell Signaling, #36406), cathepsin B (1:800, Cell Signaling, #31718), GFAP (1:100, Cell Signaling, #3670), Lamp-1 (5 µg/ml, DSHB, 1D4B), LC3B (1:200, Cell Signaling, #3868), nestin (1:100, Abcam, #ab11306), ki67 (1:200, Biocare Medical, #CRM325), neurofilament (5 µg/ml, DSHB, #2H3), OX42 (5 µg/ml, DSHB, #M1), PAX6 (5 µg/ml, DSHB, #PAX6), p62 (1:100, Abnova, #H00008878-M01), pit1 and 5-HT (1:1000, ImmunoStar, #20080). Secondary antibodies used for immunohistochemistry and immunofluorescence staining: Alexa Fluor 488 anti-mouse (1:200, Life Technology, #Z25002), Alexa Fluor 488 anti-rabbit (1:200, Life Technology, #Z25302), Cy5 anti-rat (1:200; Life Technology, #A10525), Alexa Fluor 568 anti-mouse (1:200, Life Technology, #Z25006), Alexa Fluor 568 anti-rabbit (1:200, Life Technology, #Z25306), Biotinylated Goat Anti-Rabbit (1:200, Vector Laboratories, #BA-1000), Biotinylated Horse Anti-Mouse (1:200, Vector Laboratories, BA-2000).

To determine subcellular localization of cathepsin B after NOR treatment, lysates from cytosolic compartment were extracted. Briefly, after vehicle control or NOR treatment, cells were collected by centrifugation and resuspended in cytosolic buffer (20 mM/L HEPES, 10 mM/L KCl, 1.5 mM/L MgCl$_2$, 1 mM/L EDTA, 250 mM/L sucrose, 1.5 mM/L PMSF, 1 mM/L DTT, protease inhibitor cocktail (Sigma, P8340)). The cells were lysed on ice by passing through 26 G syringe 30 times, centrifuged for 30 min at 4 °C and 13,200 rpm. The supernatant was centrifuged once again for 30 min. Protein concentration was measured using NanoDrop 2000 (ThermoFisher) and resolved by SDS-PAGE.

The following primary antibodies were used to conduct immunoblotting: Rb, p53, α/β-tubulin (1:1000, Cell Signaling, #2148), Lamp-1 (0.5 µg/ml, DSHB, 1D4B), p62 (1:1000, Abnova, # H00008878-M01), LC3B (1:1000, Cell Signaling, #3868), actin (0.5 µg/ml, DSHB, JLA20), and cathepsin B (1:1000, Cell Signaling, #31718). Secondary antibodies were: HRP-linked Anti-rabbit IgG (1:1000, Cell Signaling, #7074S), HRP-linked Anti-mouse IgG (1:1000, Cell Signaling, #7076 S) and HRP-linked Anti-rat IgG (1:1000, Cell Signaling, #7077S).

**Transmission electron microscopy (TEM)**. Primary tumor cells were treated with vehicle control or 20 µM NOR for 2 h, fixed in 2% glutaraldehyde in 0.1 M sodium cacodylate buffer (pH = 7.3) overnight. Then, cells were rinsed, post-fixed, dehydrated and infiltrated according to the procedure at The Nanoscale Biomedical Imaging Facility at The Hospital for Sick Children Research Institute, Canada. Thin sections were obtained using Leica Ultracut Ultramicrotomes and analyzed with FEI Tecnai 20 TEM.

**MRI imaging and in vivo therapeutic analysis**. Mice treated with indicated regimens were injected with gadolinium by i.v. to enhance contrast prior to MRI. Tumor volumes were calculated using Medical Image Processing, Analysis & Visualization (MIPAV) software [https://mipav.cit.nih.gov][67].

Primary tumor cells ($2 \times 10^6$) were suspended in 20 µl neurobasal media mixed 1:1 with 20 µl Matrigel and transplanted subcutaneously into the right flank of NOD/SCID mouse. Once tumors were palpable, mice were randomized and treated until end-point (tumor volume = 550 mm$^3$). Tumor volume was calculated using V = π/6 x L x W x H. Mice were treated with vehicle control, 10 mg/kg NOR, 60 mg/kg gemcitabine or a combination of both. NOR was injected i.p. 4 days a week; gemcitabine was injected i.v. once a week.

**Apoptosis assays**. Rb/p53-deleted PB cells ($1.5 \times 10^4$) were seeded in 96-well plates and treated with indicated drugs and durations. Caspase 3/7 activity was measured as per manufacturer's instructions (Promega Caspase-Glo 3/7 Assay System; #G8090).

For flow cytometry analysis, cells ($7.5 \times 10^4$) were seeded onto 24-well plates and treated with indicated drugs. 24-h post treatment, the cells were stained with PI and Annexin V (BD FITC Annexin V Apoptosis Detection Kit I; #556547) and analyzed using BD LSR Fortessa cell analyzer. FSC-A vs SSC-A plots were used to

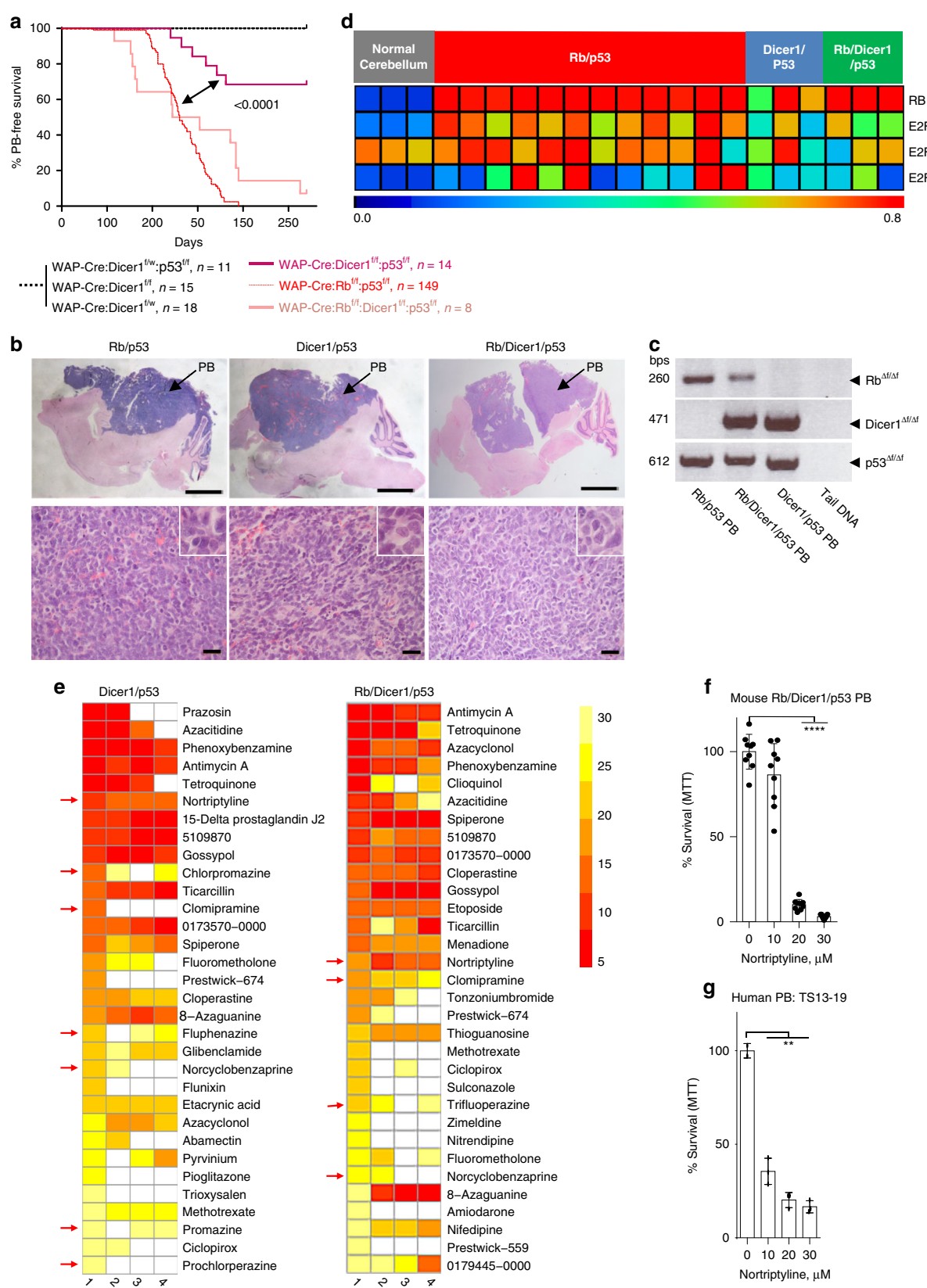

gate cells, excluding debris, as per the BD manual. Annexin V FITC vs propidium iodide (PI) plots from the gated cells uncovered populations corresponding to viable, apoptotic or necrotic cells.

**Acridine orange assays.** Primary PB cells were treated with vehicle or 30 μM NOR for 2 h. Acridine orange (final concentration = 2 μg/ml) was added 20 min

before the end of treatment. Photo-oxidation was induced by exposing cells to blue light, and changes in green intensity (eGFP) were captured using ZEISS LSM700 confocal microscope. To determine lysosome location, primary tumor cells were treated with vehicle control or 12.5 μM NOR for 2 h. Acridine orange solution was added as above and lysosomes and cytoplasm were captured with red (mCherry) and green (eGFP) filters using ZEISS LSM700 confocal microscope, respectively. Location of lysosomes relative to nucleus was assessed using ImageJ

**Fig. 8 Dicer1 plus p53 deletion via WAP-Cre induces low-penetrant pineoblastoma. a** Kaplan–Meier PB-free survival curves showing that WAP-Cre: Dicer1$^{flox/flox}$:p53$^{flox/flox}$ mice ($n = 19$, magenta) developed PB with 31.6% penetrance. Mantel-Cox test, $p < 0.0001$ vs Rb/p53-deficient PBs. WAP-Cre: Dicer1$^{flox/flox}$:Rb$^{flox/flox}$:p53$^{flox/flox}$ triple mutant mice ($n = 14$) developed pineoblastoma with reduced latency of 138 days and 92.9% penetrance. $P = 0.0003$ vs Dicer1/p53 and $P = 0.017$ vs Rb/p53-deficient PBs. **b** Representative H&E images of whole brain and histology of PBs from indicated genotypes. Inlets, high power magnification (×100) showing similar pleomorphic PBs. Scale bars, 200 μm (top), 2 μm (bottom). **c** PCR analysis confirming deletion of Rb, Dicer1 or p53 alleles in indicated PBs. **d** RB KO signature and E2F1-3 pathway activity in Rb/p53, Dicer1/p53 and Rb/Dicer1/p53 PBs relative to normal control brain. Scale bar represents pathway activity probability. For GSEA-PCA-based clustering of these lesions with human brain tumors, see Fig. 2a. **e** Connectivity mapping by GWC of mouse Dicer1/p53-deficient (left) and Rb/Dicer1/p53-deficient (right) pineoblastomas identifies multiple tricyclic drugs including NOR (red arrows). 1–4 refers to 100, 200, 300, and 500 genes per signature used for each analysis. Color scale bar represents top 30 drugs when analyzing 100 genes per signature. **f** Response of mouse Rb/Dicer1/p53-deficient pineoblastoma cells to increasing concentrations of NOR for 24 h using MTT assays. Results show values from three independent experiments each performed in triplicates. **** $P < 0.0001$. Statistical analysis by two-tailed student's $t$-test. **g** Response of human TS13-19 PB cells to increasing concentrations of NOR for 24 h by MTT assays, performed in triplicates; ** $P < 0.002$. Results are represented as mean ± SD in (**f**) and (**g**). See Source Data files for raw data.

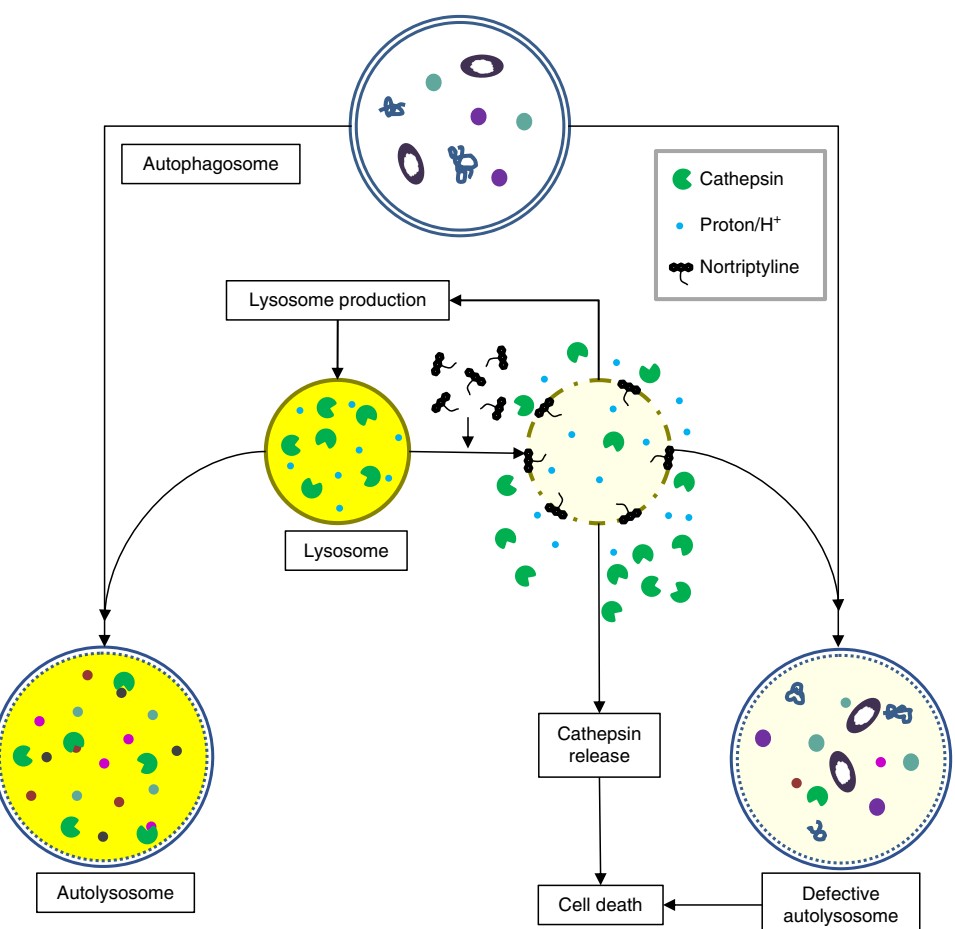

**Fig. 9 Model for the cytotoxic effect of nortriptyline on pineoblastoma.** In untreated cells (left), autophagosomes with their cargo fuse to lysosomes, which provide digestive enzymes such as cathepsins and high acidic environment (yellow) for their enzymatic activity, to form functional autolysosomes where the cargo is degraded. Both mouse and human pineoblastomas are highly sensitive to the cationic amphiphilic (CAD) tricyclic drug nortriptyline, which disrupts the lysosomes leading to release of protons and cathepsins, and to accumulation of non-functional autolysosomes that fail to degrade their payload, ultimately leading to cell demise (right). Tumor cells utilize autophagy as an escape mechanism to evade anti-neoplastic cytotoxic drugs, and, in accordance, nortriptyline effectively cooperates with gemcitabine to suppress pineoblastoma.

software. Lysosomes were considered as juxtanuclear when they were within 4 μM-range from the nucleus, and perinuclear when they were outside the 4 μM-range.

**Gene expression, pathway enrichment, and activity analyses.** To compare Rb/p53-deleted PBs and Rb/p53-mutated PBs, RNA was collected from snap-frozen tumor tissues and analyzed on Mouse Gene 1.0 ST Array (Affymetrix).

Gene expression profiles of human SHH MB, group 3 MB, and PB were analyzed using dataset from Mixed Pediatric PDX (public) – Olson – 55- MAS5.0 – u133p2 [https://hgserver1.amc.nl/cgi-bin/r2/main.cgi]. Using "view multiple genes", the expression profiles of top 10 genes from human SHH and group 3 MB were compared between human SHH MB, group 3 MB, and PB. Using "find differential expression between groups", differentially expressed genes between human SHH MBs and PBs were identified. From these, 95 genes were used to determine the expression profile in Rb/p53-deleted PBs.

GSEA was performed with v2 using PreRanked Method, a two-sided calculation. The resulting enrichment scores (ES) were normalized by multiple sample correction for normalized enrichment score (NES). Furthermore, the multiple gene/sample correction was done with 1000 permutations to generate FDR values. Specific values for metastatic pathways: WINNEPENNINCKX_ MELANOMA_METASTASIS_UP, Nominal $p$-value: <0.0001 FDR: 0.0302 ES: 0.3505 Normalized ES: 1.6770; TOYOTA_TARGETS_OF_MIR34B_AND_MIR34C, Nominal $p$-value: <0.0001 FDR: 0.0032 ES: 0.3660 Normalized ES: 1.9651; RICKMAN_METASTASIS_UP, Nominal $p$-value: <0.0001 FDR: 0.0012 ES: 0.3946 Normalized ES: 2.0646 BIDUS_METASTASIS_UP, Nominal $p$-value: <0.0001 FDR: 0 ES: 0.4592 Normalized ES: 2.2868.

Dicer1-deficient tumors were analyzed using Mouse Gene 2.0 ST Array. For comparative GSEA-PC and pathway activity analyses we used the specific probes in each platform (1.0 ST and 2.0 ST). Pathway activity was calculated for RB KO signature and E2F1, E2F2, and E2F3 pathways using binary regression models' script Binreg v2.0 in the MatLab environment and related pathway training data set as described[68].

**In silico drug prediction**. Transcriptional profiles of drug-treated cancer cell lines from the BROAD Connectivity map initiative (CMAP, $n = 1309$ drugs) were obtained using the PharmacoGx package and a linear regression model as described in R[35]. Pre-computed CMAP drug perturbation signatures for 1288 drugs were used in the downstream analysis. These signatures signify the drug concentration effect on the transcriptional state of the cell, and were used to identify genes whose expression is perturbed by drug treatment. Gene expression 'signatures' pertaining to each of the mouse models were obtained by comparing samples of each model against normal tissue samples, to identify differentially expressed genes using the limma package[69]. A linear model per gene was fit using the lmfit function, followed by an empirical Bayes adjustment using the eBayes function to generate several statistics for differential expression (t-stat, log-odds ratio). Final annotations and multiple-testing correction (FDR) adjustments were taken using the topTable function, and genes were considered based on a threshold of FDR < 0.01.

Drug repurposing of CMAP against each signature was conducted on the orthologous genes common between the mouse signatures and CMAP perturbation signatures. Rb/p53 'signatures' specific to each mouse model were chosen by selecting equal numbers of significant up- and downregulated differentially expressed genes. Several signature sizes were tested, spanning 100, 300, and 500 genes, respectively. Rb/p53 mouse signatures were compared against drug perturbation signatures from CMAP to identify drugs that could reverse the mouse model signature (i.e., could be a potential therapeutic drug). Connectivity scores between CMAP drug perturbation signatures and each of the mouse model signatures were computed using the connectivityScore function of the PharmacoGx package. Connectivity scores were computed using the GSEA method, based on the Kolmogorov-Smirnov (KS) statistic, as well as the GWC method (based on the weighted spearman statistic). Drugs were ranked by their connectivity score and associated p-value, with more negative connectivity scores indicating the ability for a given drug to reverse the mouse signature. Top hits were considered by filtering drugs with connectivity scores below −0.5 for both GSEA-based and GWC-based analyses. Similar conditions were used for CMAP analysis of mouse Dicer1/p53- and Rb/Dicer1/p53-deficient PBs.

**GSEA-PCA**. Human and mouse gene expression profiles were collated from in-house experiments, and datasets from the NCBI Gene Expression Omnibus (GEO) including MB (GSE37382), GBM (GSE36245), and Rentioblastoma (GSE29685, GSE24673, GSE59983). Gene expression profiles of each dataset were processed separately within R (version 3.5.0) and Bioconductor, using the Geoquery and affy packages and probe level information from BrainArray, as described[22]. Single-sample Gene Set Enrichment Analysis (ssGSEA)[70] was conducted on human and mouse expression profiles using the GSVA package (version 1.32.0) in R. Resultant gene sets were filtered to encompass only those genesets common to both mouse and human datasets. Single-sample geneset ES across the genesets were then combined to generate an ssGSEA-ranked matrix. Principal component analysis (PCA) was then conducted on 578 ssGSEA-ranked genes sets (rank matrix). Genome-wide connectivity scores (GWC method) were determined using a weighted spearman correlation statistic. In supplementary Tables, significance of the scores was determined by permutation testing (1000 permutations) and a two-sided test. Connectivity scores generated using the GSEA method are based on KS statistic implemented in the piano package. Returned $p$-values are distinct-directional p-values adjusted following permutation sampling (1000 permutations).

**Statistics and reproducibility**. Data are presented as mean ± SD. All statistical analyses listed in figure legends and Kaplan–Meier survival curves were generated using GraphPad Prism. Quantification of fluorescent intensity was performed using ImageJ software. With the exception of human PB TS13-19 cells, which were analyzed once in triplicates as they were extremely difficult to propagate, all drug studies were performed in three or more replicates each, in three biological replicates.

**Reporting summary**. Further information on research design is available in the Nature Research Reporting Summary linked to this article.

## Data availability

Accession number: The NCBI Gene Expression Omnibus accession number for the microarray results reported in this paper is GSE124537. GSE37382, GSE36245, GSE29685, GSE24673 and GSE59983 are publicly available.

Source data: The source data underlying Figs. 1e; 3g; 4c–e; 5a–e; 6a–b, e, f; 7a–d, 8c, f–g; and Supplementary Figs. 5a, 6a–b, d; 7; 8a, 9a–d; are provided as a Source Data file.

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

## Acknowledgements

We thank Dr. Michael Salter for access to his freezing microtome. This research was funded by grants from the Canadian Institute of Health Research (CIHR) to E.Z. and from Terry Fox Research Institute Program Grant to M.D.T. and E.Z.

## Author contributions

P.E.D.C. performed experiments, analyzed data and wrote the manuscript; R.G.A. and J.T. performed experiments; D.M.A.G., J.C.L., D.Y.W., B.L., A.D., M.R., B.H., and D.S. conducted bioinformatic analysis; X.X. helped with flow cytometry analysis, S.G.K. provided an established human pineoblastoma cell line; Z.J., L.G., Y.B.D., S.C., B.H.K., A.H., and M.D.T. advised and supervised the analysis; E.Z. supervised and coordinated the analysis, analyzed results and wrote the manuscript.

## Competing interests

The authors declare no competing interests.
