## [Peer Review File · Nature Communications]

Reviewers' Comments:

Reviewer #1:

Remarks to the Author:

The authors report that inactivation of Rb1/DICER1 plus p53 via WAP-Cre transgene leads to metastatic pineoblastomas in immune-competent mice. The resultant tumors have short latency and high penetrance, also resemble human disease, supported by histology, imaging and cluster analysis. The authors further showed Rb/p53-mutated mice harboring p53 R270H stabilizing mutation would develop pineoblastomas as Rb/p53-deficient ones do, but with higher metastatic potential. To seek treatment modality, the authors used in silico analysis and rediscovered that nortriptyline (NOR) can inhibit autophagy. This agent suppressed tumor growth both in vitro and in vivo (flank only) on Rb/p53-deficient pineoblastoma models. NOR could also synergize with gemcitabine, curiously improving survival in animals with flank tumors, despite only modestly impacting tumor burden.

Is WAP1 expression in the pineal physiological or is this an artefact of the Cre driver used? This issue should be addressed.

While it is long known that p53 and Rb loss leads to pineoblastoma (PMID: 12096340 acknowledged by authors and PMID: 7951317, a high-impact paper 25 years ago, curiously not cited here), this study established a Rb/p53-deficient model with full penetrance, showed that p53 stabilizing mutation lead to a more metastatic phenotype (though p53 not typically found in human disease), also addressed the tumor's resemblance to human disease. The potential medications found could be of interest to the research community if shown to work in an orthotopic setting, results extended to human PDX models also in an orthotopic setting, and if mechanistic insights could be clarified and extended to treated tumors in-vivo.

WAP-Cre Rb/p53-deficient and Rb/p53-mutated mice all develop pineoblastomas with high penetrance. It is a bit confusing that at the end of the results, the authors showed DICER1/p53-deleted mice developed pineoblastoma, but no genetic data was shown. This genetic analysis should be added. Additionally, can the authors address the need for and relevance of p53 mutation? Is this mutation ever found in human tumors?

While some previously established mechanisms are confirmed for NOR's mechanism of action against pineoblastoma, how NOR synergizes with gemcitabine (GEM) is not explored. This analysis should be added. The authors described combination of NOR and GEM provides significant survival benefit and inhibits tumor growth in subcutaneous models, but GEM plus NOR has a quite modest effect on tumor volume compared to each used alone in the Rb/p53-deficient models. Mice typically tolerate large flank tumors, as evidenced by the bar graph in Fig 7e. Can the authors explain why such a modest difference in volume led to such a dramatic difference in survival? And can these results be corroborated orthotopically in this and in human PDX models.

Reviewer #2:

Remarks to the Author:

This manuscript, submitted by Chung and colleagues reports on an analysis of mice engineered to develop pineoblastoma via tissue specific deletion of Rb or Dicer combined with deletion/mutation of p53. The authors find that Rb + p53 mutation = aggressive highly penetrant tumors; Dicer + p53 deletion = lower penetrance. Using pathway enrichment analysis of RNAseq data generated from pineoblastomas from the Rb/p53 mouse models, the authors identify tricyclic antidepressants (TCA) as a therapeutic candidate. They identify nortriptyline as the most potent of the TCAs they tested and go on to show using chemical biological approaches that it is nortriptyline's effect on the lysosome and autophagy that accounts for the anti-pineoblastoma effect rather than nortriptyline's role as an SSRI.

Overall the authors provide very nice data that will be of keen interest to the neuro-oncology community and provides some pre-clinical rationale for autophagy inhibitors as a therapeutic strategy in patients with pineoblastoma. However, I found this manuscript very difficult to read, especially towards the end of the manuscript where it seemed to become a mish mash of results rather than a tight arc. Part of this was due to the placement of the Dicer/p53 mouse model at the end of the manuscript rather than incorporating (and thoroughly analyzing this model) with the Rb/p53 pineoblastoma model. I personally think the Dicer/p53 results could be left out, which would result in a more tightly woven story of how analysis of the Rb/p53 pineoblastomas informed the discovery of TCAs that led to the finding that autophagy as an important component to pineoblastoma growth and survival. Along these lines, the authors did not extensively work up the Dicer/p53 pineoblastomas. I would expect the authors to perform the repertoire of studies such as gene expression profiling, GSEA, CMAP analysis, TCA treatment / autophagy analysis that they did with the Rb/p53 pineoblastomas and compare/contrast the two.

My only other major concern is that very high doses of nortriptyline were required to see an in vitro response, which was quite discrepant to the in vivo doses used in the survival analysis. The authors should address this discrepancy.

Minor

Why not pick another candidate from the CMAP screen instead of extrapolate data from a completely different histological subtype of tumor (medulloblastoma)?

Why do the drug in vivo (and in vitro) efficacy studies in the p53-deleted strain instead of the more aggressive p53-mutated strain?

REBUTTEL

Reviewer #1 (Remarks to the Author):

The authors report that inactivation of Rb1/DICER1 plus p53 via WAP-Cre transgene leads to metastatic pineoblastomas in immune-competent mice. The resultant tumors have short latency and high penetrance, also resemble human disease, supported by histology, imaging and cluster analysis. The authors further showed Rb/p53-mutated mice harboring p53 R270H stabilizing mutation would develop pineoblastomas as Rb/p53-deficient ones do, but with higher metastatic potential. To seek treatment modality, the authors used in silico analysis and rediscovered that nortriptyline (NOR) can inhibit autophagy. This agent suppressed tumor growth both in vitro and in vivo (flank only) on Rb/p53-deficient pineoblastoma models. NOR could also synergize with gemcitabine, curiously improving survival in animals with flank tumors, despite only modestly impacting tumor burden.

Response: First, regarding “This agent suppressed tumor growth both in vitro and in vivo (flank only) on Rb/p53-deficient pineoblastoma models.” - we would like to point out that in fact we demonstrated by MRI that the agent, nortriptyline, effectively suppresses pineoblastoma not only in flank assays but also and more importantly in our preclinical model (**Figures 4f and 7d**).

Second, with regards to the combination therapy –% survival in the flank assay in **Fig. 7e** is defined on page 15 (Methods), as tumor volume over 550mm³. We have added this definition of % survival in the revised legend for **Fig. 7e**. Such endpoint (rather than death) is commonly used, e.g. Clin Cancer Res. 2015 Dec 15;21(24):5488-98. (PMID: 26169967).

Is WAP1 expression in the pineal physiological or is this an artefact of the Cre driver used? This issue should be addressed.

Response: We analyzed WAP gene expression in several data sets (only two of which are shown in the revised manuscript but same results are seen in all datasets). There is very low WAP expression (likely background levels) in the pineal gland compared to 51 other CNS regions which are also extremely low (new **supplemental Fig. 2**). We also compared WAP expression in the CNS to the mammary gland in various developmental stages, showing robust WAP expression during pregnancy. Notably, our WAP-Cre mT/mG reporter analysis identified a wide expression of green-positive cells in the pineal gland but very rare GFP+ cells in the CNS. Together, these results indicate, as the Reviewer suggested, that expression of the WAP-Cre transgene in the pineal gland is likely an artefact of the Cre driver transgene. We describe these results on **page 3** in the text, and **page 10** in Discussion.

While it is long known that p53 and Rb loss leads to pineoblastoma (PMID: 12096340 acknowledged by authors and PMID: 7951317, a high-impact paper 25 years ago, curiously not cited here), this study established a Rb/p53-deficient model with full

penetrance, showed that p53 stabilizing mutation lead to a more metastatic phenotype (though p53 not typically found in human disease), also addressed the tumor's resemblance to human disease.

Response: We referenced a paper showing tissue-specific knockout of Rb and p53 with an IRBP-Cre transgene, which only induces a partial penetrance for PB as well as multiple other brain lesions. As requested, we have now referenced PMID: 7951317, which describes cooperation of germline mutations in Rb and p53 (Nat Genet. 1994 Aug;7(4):480-4) - **page 2**.

The potential medications found could be of interest to the research community if shown to work in an orthotopic setting, results extended to human PDX models also in an orthotopic setting, and if mechanistic insights could be clarified and extended to treated tumors in-vivo.

Response: We agree with the Reviewer that despite their caveats (instability, immune-compromised host), analysis of human PB PDX models would strengthen our results. Unfortunately, PB cells are notoriously difficult to propagate *in vitro* and *in vivo*. Indeed, only one PB cell line has been reported in the literature and is very difficult to culture. We show in the manuscript that this line, obtained from Seok-Gu Kang (co-author), is sensitive to Nortriptyline *in vitro* (Fig. 4d). We also show that Nortriptyline cooperates with gemcitabine to kill a Group 3 medulloblastoma cell line *in vitro* – though the synergism was not as good as with PB cells.

Two (untagged) PB PDXs are commercially available (<https://research.fhcr.org/olson/en/btrl.html>). However, it takes 6 months for these PDXs to develop pineoblastoma following intracranial transplantation. Perhaps multiple or continuous passages plus tagging with a luciferase reporter would allow the use these PDXs in future studies to assess potential therapies, but right now this is not feasible. In addition, the status of RB1 in these two PDXs is not known (they are not derived from trilateral familial RB, and therefore unlikely to have RB1 mutation/deletion). We discussed this limitation to the study (lack of PDX analysis) on **page 12**.

WAP-Cre Rb/p53-deficient and Rb/p53-mutated mice all develop pineoblastomas with high penetrance. It is a bit confusing that at the end of the results, the authors showed DICER1/p53-deleted mice developed pineoblastoma, but no genetic data was shown. This genetic analysis should be added. Additionally, can the authors address the need for and relevance of p53 mutation? Is this mutation ever found in human tumors?

Response: We presented WAP-Cre:Dicer1^{fl/fl} mice to demonstrate that this deleter line can be used to generate other models of Pineoblastoma. Our observation that WAP-Cre mediated deletion of Dicer1 induces PB suggests that it targets a common cell(s) of origin for both RB1 and DICER1 germline Pineoblastoma. In the revised manuscript, we have expanded our colonies of Dicer1/p53 and triple Rb/Dicer1/p53 knockout mice and provided better Kaplan-Meier PB-free curves in comparison to Rb/p53 lesions (new **Fig. 8a**). We show that WAP-Cre:Dicer1^{fl/fl};p53^{fl/fl} have incomplete penetrance and delayed

latency compared with the Rb/p53 models, whereas WAP-Cre:Rb/Dicer1/p53 triple knockout mice show shorter latency than Rb/p53 mice and 100% penetrance.

In addition, we subjected these double WAP-Cre:Dicer1^{fl/fl};p53^{fl/fl} and triple WAP-Cre:Rb^{fl/fl};Dicer1^{fl/fl};p53^{fl/fl} PBs to mRNA microarray analysis and performed two major bioinformatic analysis:

- (i) Pathway activity analysis of RB knockout as well as E2F1, E2F2 and E2F3 signalling on all our mouse models. This analysis revealed that Rb/p53 and Rb/Dicer1/p53-deletion PBs exhibit high RB KO signature loss compared to control brain tissue (new **Fig. 8d**). One Dicer1/p53 PB showed similar increase in RB loss signalling, whereas the other two tumors showed elevated but not robust RB loss activity. Consistent with this, E2F1-3 pathway activation analysis revealed elevated signalling in one or more of these transcription factors downstream of RB-loss in each Dicer1/p53 lesion. There was high activity of E2F2 in the single PB with robust RB loss signalling, which may reflect an oncogenic alteration in this particular tumor.
- (ii) GSEA-PC analysis showing, importantly, that mouse Dicer1/p53 and Rb/Dicer1/p53 PBs cluster with mouse Rb/p53 and human PBs as well as with mouse retinoblastoma but not GBM (new **Fig 2a, Supplementary Fig 3**).

Regarding p53 mutation: first, Rb+/- mice only develop pituitary tumors, not retinoblastoma. Targeted loss of Rb alone in retina progenitor cells also fail to induce retinoblastoma. Co-deletion of Rb together with p107, p130 or p21 is required to induce retinoblastoma. Thus, unlike humans where RB1 mutant carriers develop bilateral retinoblastoma with near 100% penetrance, mouse retinal cells need a reduced CDK inhibitor background to undergo neoplastic conversion. In other tissues in the mouse (breast, prostate, ovary), combined loss of Rb plus p53 is necessary to induce cancer. In aggressive forms of human cancers, e.g. triple negative breast cancer, RB1 and TP53 are often disrupted together. Thus, it is not surprising that one needs to inactivate Rb together with p53 to induce tumorigenesis in the pineal gland.

Second, as noted in the manuscript, Saab, R., et al. (Cancer Res 69, 440-448 (2009)) reported on a TP53 stabilizing mutation in a highly metastatic PB patient. Thus, our models mimic most aggressive forms of PB.

While some previously established mechanisms are confirmed for NOR's mechanism of action against pineoblastoma, how NOR synergizes with gemcitabine (GEM) is not explored. This analysis should be added. The authors described combination of NOR and GEM provides significant survival benefit and inhibits tumor growth in subcutaneous models, but GEM plus NOR has a quite modest effect on tumor volume compared to each used alone in the Rb/p53-deficient models. Mice typically tolerate large flank tumors, as evidenced by the bar graph in Fig 7e. Can the authors explain why such a modest difference in volume led to such a dramatic difference in survival? And can these results be corroborated orthotopically in this and in human PDX models.

Response: As discussed above, the survival curves in the flank assays were not based on death as end point but as tumor volume of 550 mm³. The MRI on preclinical models are difficult and expensive experiments, and therefore rather than repeating the MRI analysis with a larger group of mice, we opted to use flank assays which allow multiple mice to be analyzed at the same time following transplantation. The MRI analysis does show a trend toward synergy between GEM and NOR, which is confirmed in the flank assays.

Re Xenograft assays – please see response to a similar question above regarding the current infeasibility of these experiments.

Finally, as requested, we tested for cooperation between NOR and GEM in the induction of cell death by Annexin V/PI flow cytometric analysis. The results shown in new **Supplementary Fig. 9c** reveal that these drugs cooperate to promote necrotic or late apoptotic AnnexinV⁺PI⁺ cell death after 24 hr treatment.

Reviewer #2 (Remarks to the Author):

This manuscript, submitted by Chung and colleagues reports on an analysis of mice engineered to develop pineoblastoma via tissue specific deletion of Rb or Dicer combined with deletion/mutation of p53. The authors find that Rb + p53 mutation = aggressive highly penetrant tumors; Dicer + p53 deletion = lower penetrance. Using pathway enrichment analysis of RNAseq data generated from pineoblastomas from the Rb/p53 mouse models, the authors identify tricyclic antidepressants (TCA) as a therapeutic candidate. They identify nortriptyline as the most potent of the TCAs they tested and go on to show using chemical biological approaches that it is nortriptyline's effect on the lysosome and autophagy that accounts for the anti-pineoblastoma effect rather than nortriptyline's role as an SSRI.

Overall the authors provide very nice data that will be of keen interest to the neuro-oncology community and provides some pre-clinical rationale for autophagy inhibitors as a therapeutic strategy in patients with pineoblastoma. However, I found this manuscript very difficult to read, especially towards the end of the manuscript where it seemed to become a mish mash of results rather than a tight arc. Part of this was due to the placement of the Dicer/p53 mouse model at the end of the manuscript rather than incorporating (and thoroughly analyzing this model) with the Rb/p53 pineoblastoma model. I personally think the Dicer/p53 results could be left out, which would result in a more tightly woven story of how analysis of the Rb/p53 pineoblastomas informed the discovery of TCAs that led to the finding that autophagy as an important component to pineoblastoma growth and survival. Along these lines, the authors did not extensively work up the Dicer/p53 pineoblastomas. I would expect the authors to perform the repertoire of studies such as gene expression profiling, GSEA, CMAP analysis, TCA treatment / autophagy analysis that they did with the Rb/p53 pineoblastomas and compare/contrast the two.

Response: We thank the Reviewer for the kind words. We have now revised the text describing the Dicer 1 model in the end of the Result section as requested, and tried to further simplify the entire manuscript.

With regards to the Dicer1 model – we think that its inclusion is important for two reasons: First, it demonstrates that the WAP-Cre deleter line can be used to generate other models of Pineoblastoma. Second, our observation that WAP-Cre mediated deletion of Dicer1 induces PB suggests that this transgene targets a common cell(s) of origin for both RB1 and DICER1 germline Pineoblastoma. In the revised manuscript, we have expanded our colonies of Dicer1/p53 and triple Rb/Dicer1/p53 knockout mice and provided better Kaplan-Meier PB-free curves in comparison to Rb/p53 lesions (new **Fig. 8a**). We show that WAP-Cre:Dicer1^{fl/fl}:p53^{fl/fl} have incomplete penetrance and delayed latency compared with the Rb/p53 models, whereas WAP-Cre:Rb/Dicer1/p53 triple knockout mice show shorter latency than Rb/p53 mice and 100% penetrance.

In addition, we subjected these double WAP-Cre:Dicer1^{fl/fl}:p53^{fl/fl} and triple WAP-Cre:Rb^{fl/fl}:Dicer1^{fl/fl}:p53^{fl/fl} PBs to mRNA microarray analysis and performed two major bioinformatic analysis:

- (i) Pathway activity analysis of RB knockout as well as E2F1, E2F2 and E2F3 signalling on all our mouse models. This analysis revealed that Rb/p53 and Rb/Dicer1/p53-deletion PBs exhibit high RB KO signature loss compared to control brain tissue (new **Fig. 8d**). One Dicer1/p53 PB showed similar increase in RB loss signalling, whereas the other two tumors showed elevated but not robust RB loss activity. Consistent with this, E2F1-3 pathway activation analysis revealed elevated signalling in one or more of these transcription factors downstream of RB-loss in each Dicer1/p53 lesion. There was high activity of E2F2 in the single PB with robust RB loss signalling, which may reflect an oncogenic alteration in this particular tumor.
- (ii) GSEA-PC analysis showing, importantly, that mouse Dicer1/p53 and Rb/Dicer1/p53 PBs cluster with mouse Rb/p53 and human PBs as well as with mouse retinoblastoma but not GBM (new **Fig 2a, Supplementary Fig 3**).

My only other major concern is that very high doses of nortriptyline were required to see an *in vitro* response, which was quite discrepant to the *in vivo* doses used in the survival analysis. The authors should address this discrepancy.

Response: For *in vitro* experiments, assays were performed after short periods of treatment, e.g. one day for MTT assays. In contrast, *in vivo* therapies were performed over extended periods allowing accumulative effects.

Nortriptyline is metabolized in the liver by demethylation and hydroxylation as well as by conjugation with glucuronic acid. The most active metabolite is 10-hydroxynortriptyline, which has the same pharmacological profile as nortriptyline, and is most abundant in the plasma. This or other derivatives may be more active than nortriptyline in disrupting the lysosome, but additional analysis is required to examine this possibility.

Notably, our *in vivo* assays were performed with doses that are clinically attainable. Briefly, NOR doses in humans are ~150 mg/day, which for a 75Kg average person = 2 mg/Kg. Small animals such as mice metabolize drugs faster than human (large animals). Indeed the dose for mice can be as high 12.3 fold higher than humans (see PMID: 27057123). Thus, 2mg/kg x 12.3 = 24.6 mg/kg for mice. For NOR treatment alone, we started with 20mg/kg and slowly increased the dose to 40 mg/kg, which is double the dose in human but with no side effects. For combination with GEM, we kept the dose of NOR at 20 mg/kg for mouse models or used 10mg/kg for the flank assays. We added the reference above and discussed these dosing considerations in the revised manuscript (Discussion).

Minor

Why not pick another candidate from the CMAP screen instead of extrapolate data from a completely different histological subtype of tumor (medulloblastoma)?

Response: Our observation that NOR blocks autophagy prompted us to test for synergy with an antineoplastic drug as it is well-established that many drugs induce autophagy, which allows tumor cells to escape cytotoxic drugs and survive. As gemcitabine is an FDA-approved antineoplastic drug that showed efficacy against G3M in preclinical models, and given the similarity of PB to G3M, and that our initial candidate screen revealed that mouse Rb/p53 PB are extremely sensitive to this drug, it was a logical step to test for synergy between NOR and GEM.

Why do the drug in vivo (and in vitro) efficacy studies in the p53-deleted strain instead of the more aggressive p53-mutated strain?

Response: For analysis of metastasis, the Rb/p53-mutant mice would indeed be superior. However, for primary PB, we showed that the kinetic of tumor formation for the two models (Rb/p53-deletion vs Rb/p53-mut) is virtually identical, and there is therefore no difference between the two strains. The reason for using the Rb/p53 deletion was practical; we initiated the study with this line and generated Rb/p53-mut mice only later on. We had a large colony of Rb/p53-deletion mice that allowed the drug treatment assays, which necessitates multiple breeding cages to obtain young mice and controls with the right genotype.

**

Reviewers' Comments:

Reviewer #1:

Remarks to the Author:

Revised manuscript adequately addresses my prior critique

Reviewer #2:

Remarks to the Author:

I appreciate the authors' consideration for generating additional data on the Dicer/p53 mouse model. However, I was disappointed to see that their analysis of the gene expression data from these Dicer/p53 pineoblastomas was incomplete. In particular (unless I've missed it somewhere), there is a glaring omission of a simple CMAP analysis on these expression data. As this is essentially just another GSEA run using these specific genesets, it should take an afternoon to do (or even less time). This is important to substantiate whether autophagy inhibitors are universally effective against pineoblastoma versus being specific to Rb-deleted/mutated pineoblastoma.

I also gather that pineoblastomas from the Dicer/p53 or Dicer/Rb/p53 mice were attempted to be cultured similar to the Rb/p53 tumors? And if this was the case, a simple set of dose-response curves would be expected to fully incorporate this model as part of this paper.

Regardless of the outcome of the above experiments, as both positive and negative results would prove informative to readership, I feel if the authors fulfill these two recommended experiments that it should warrant publication of this manuscript.

REBUTTEL

Reviewers' comments:

Reviewer #1 (Remarks to the Author):

Revised manuscript adequately addresses my prior critique

Reviewer #2 (Remarks to the Author):

I appreciate the authors' consideration for generating additional data on the Dicer/p53 mouse model. However, I was disappointed to see that their analysis of the gene expression data from these Dicer/p53 pineoblastomas was incomplete. In particular (unless I've missed it somewhere), there is a glaring omission of a simple CMAP analysis on these expression data. As this is essentially just another GSEA run using these specific genesets, it should take an afternoon to do (or even less time). This is important to substantiate whether autophagy inhibitors are universally effective against pineoblastoma versus being specific to Rb-deleted/mutated pineoblastoma.

I also gather that pineoblastomas from the Dicer/p53 or Dicer/Rb/p53 mice were attempted to be cultured similar to the Rb/p53 tumors? And if this was the case, a simple set of dose-response curves would be expected to fully incorporate this model as part of this paper.

Regardless of the outcome of the above experiments, as both positive and negative results would prove informative to readership, I feel if the authors fulfill these two recommended experiments that it should warrant publication of this manuscript.

Response: As requested by the Reviewer, we have performed CMAP analysis on the Dicer1/p53 and Dicer1/Rb/p53 pineoblastoma models. Strikingly, we found that just like with the Rb/p53-deficient PBs, nortriptyline together with other tricyclic compounds is predicted to target Dicer1/p53 and, to a lesser extent, Dicer1/Rb/p53 pineoblastomas.

We were so far unable to establish primary tumor cells from Dicer1/p53 pineoblastomas (compared to 50% success rate for Rb/p53 and 33% for Rb/Dicer1/p53). Due to the low penetrance of these lesions, and because the Reviewers did not request such *in vitro* drug response analysis in the initial Review, we have not established primary cells from this mouse model. However, we were successful in growing primary cultures from two independent Dicer1/Rb/p53 pineoblastomas and have now determined their sensitivity to nortriptyline. We found that these cells are highly sensitive to nortriptyline (new Fig 8f). In revised Fig. 8g, we show that nortriptyline efficiently suppresses growth of the only available human pineoblastoma cell line, TS13-19.

Together, our results demonstrate that diverse mouse and human pineoblastomas are sensitive to nortriptyline/lysosome disruption, pointing to this approach as even more viable than we previously thought. We are grateful to the Reviewer for requesting this additional analysis.

In addition, we updated the Kaplan-Meier PB-free survival curve (revised Fig. 8a). We also referenced a recent (December, 2019) manuscript by our clinical scientist co-author (A. Huang) on the classification of human pineoblastomas (reference #13, page 3 and Discussion). This paper shows that DICER1 pineoblastomas occur in older children and have better prognosis than RB1 pineoblastomas, which is exactly what we demonstrate in our mouse models (Fig. 8a).